# C-type natriuretic peptide facilitates autonomic Ca²⁺ entry in growth plate chondrocytes for stimulating bone growth

Yuu Miyazaki[1†], Atsuhiko Ichimura[1†], Ryo Kitayama[1], Naoki Okamoto[1], Tomoki Yasue[1], Feng Liu[1], Takaaki Kawabe[1], Hiroki Nagatomo[1], Yohei Ueda[2], Ichiro Yamauchi[2], Takuro Hakata[2], Kazumasa Nakao[2], Sho Kakizawa[1], Miyuki Nishi[1], Yasuo Mori[3], Haruhiko Akiyama[4], Kazuwa Nakao[5], Hiroshi Takeshima[1]*

[1]Graduate School of Pharmaceutical Sciences, Kyoto University, Kyoto, Japan; [2]Graduate School of Medicine, Kyoto University, Kyoto, Japan; [3]Graduate School of Engineering, Kyoto University, Kyoto, Japan; [4]Graduate School of Medicine, Gifu University, Gifu, Japan; [5]Medical Innovation Center, Kyoto University, Kyoto, Japan

*For correspondence: takeshima.hiroshi.8m@kyoto-u.ac.jp

†These authors contributed equally to this work

Competing interest: The authors declare that no competing interests exist.

**Abstract** The growth plates are cartilage tissues found at both ends of developing bones, and vital proliferation and differentiation of growth plate chondrocytes are primarily responsible for bone growth. C-type natriuretic peptide (CNP) stimulates bone growth by activating natriuretic peptide receptor 2 (NPR2) which is equipped with guanylate cyclase on the cytoplasmic side, but its signaling pathway is unclear in growth plate chondrocytes. We previously reported that transient receptor potential melastatin-like 7 (TRPM7) channels mediate intermissive Ca²⁺ influx in growth plate chondrocytes, leading to activation of Ca²⁺/calmodulin-dependent protein kinase II (CaMKII) for promoting bone growth. In this report, we provide evidence from experiments using mutant mice, indicating a functional link between CNP and TRPM7 channels. Our pharmacological data suggest that CNP-evoked NPR2 activation elevates cellular cGMP content and stimulates big-conductance Ca²⁺-dependent K⁺ (BK) channels as a substrate for cGMP-dependent protein kinase (PKG). BK channel-induced hyperpolarization likely enhances the driving force of TRPM7-mediated Ca²⁺ entry and seems to accordingly activate CaMKII. Indeed, ex vivo organ culture analysis indicates that CNP-facilitated bone growth is abolished by chondrocyte-specific *Trpm7* gene ablation. The defined CNP signaling pathway, the NPR2-PKG-BK channel–TRPM7 channel–CaMKII axis, likely pinpoints promising target proteins for developing new therapeutic treatments for divergent growth disorders.

## Editor's evaluation

With the new additional data and descriptions, the paper in its current state is well organized and data presented add a new information on the role of C-type natriuretic peptide and how it facilitates autonomic Ca²⁺ entry in chondrocytes and modulates bone growth.

## Introduction

The development of skeletal long bones occurs through endochondral ossification processes, during which chondrocyte layers form the growth plates at both ends of bone rudiments, and then the

expanded cartilage portions are gradually replaced by trabecular bones through the action of osteo-clasts and osteoblasts (*Berendsen and Olsen, 2015*). Therefore, bone size largely depends on the proliferation of growth plate chondrocytes during endochondral development. On the other hand, atrial (ANP), brain (BNP), and C-type (CNP) natriuretic peptides regulate diverse cellular functions by activating the receptor guanylate cyclases, NPR1 and NPR2 (*Nakao et al., 1996*). Of the natriuretic peptides, CNP exclusively stimulates bone development by acting on growth plate chondrocytes expressing the CNP-specific receptor NPR2 (*Nakao et al., 1996*; *Wit and Camacho-Hübner, 2011*; *Peake et al., 2014*). Indeed, loss- and gain-of-function mutations in the human *NPR2* gene cause acromesomelic dysplasia and skeletal overgrowth disorder, respectively (*Vasques et al., 2014*; *Wit et al., 2016*). Furthermore, translational studies have been probing the benefits of CNP treatments in various animal models with impaired skeletal growth, and a phase III clinical trial of CNP therapy has recently been completed and approved for treatment of achondroplasia patients primarily resulting from mutations in the *FGFR3* gene (*Savarirayan et al., 2020*). It is thus likely that NPR2 guanylate cyclase controls chondrocytic cGMP content during growth plate development. Downstream of NPR2 activation, cGMP-dependent protein kinase (PKG) seems to phosphorylate target proteins to facili-tate growth plate chondrogenesis (*Peake et al., 2014*). Activated PKG is postulated to stimulate the biosynthesis of growth plate extracellular matrix by playing an inhibitory role in the mitogen-activated protein kinase Raf–MEK–ERK cascade (*Krejci et al., 2005*). In parallel, glycogen synthase kinase 3β (GSK3β) is likely activated by PKG-mediated phosphorylation, leading to the hypertrophic maturation of growth plate chondrocytes (*Kawasaki et al., 2008*). However, it is still unclear how CNP promotes bone growth at the molecular level, and it is important to further address CNP signaling cascade in growth plate chondrocytes.

In the transient receptor potential channel superfamily, the melastatin subfamily member 7 (TRPM7) forms a mono- and divalent cation-permeable channel in various cell types and participates in important cellular processes including cell growth and adhesion (*Fleig and Chubanov, 2014*). We recently reported that growth plate chondrocytes generate autonomic intracellular $Ca^{2+}$ fluctuations, which are generated by the intermittent gating of TRPM7 channels, and also that TRPM7-mediated $Ca^{2+}$ entry activates $Ca^{2+}$/calmodulin-dependent protein kinase II (CaMKII), facilitating endochondral bone growth (*Qian et al., 2019*). Based on these observations, we explored the link between CNP signaling and TRPM7-mediated $Ca^{2+}$ entry through the experiments described in this report. Our data obtained clearly indicate that big-conductance $Ca^{2+}$-dependent $K^+$ (BK) channels play a key role in the functional coupling between NPR2 and TRPM7 channels in growth plate chondrocytes.

## Results
### CNP facilitates spontaneous $Ca^{2+}$ fluctuations in growth plate chondrocytes

In the growth plates of developing bones, proliferating cartilage cells, designated as round and columnar chondrocytes, frequently exhibit weak increases and decreases in intracellular $Ca^{2+}$ concen-tration under resting conditions (*Qian et al., 2019*). On the other hand, previous in vivo studies demonstrated that CNP application (>1 μmol/kg) stimulates endochondral bone growth (*Nakao et al., 1996*). In our Fura-2 imaging of round chondrocytes within femoral bone slices prepared from wild-type mice, CNP pretreatments (30–300 nM for 1 hr) dose-dependently facilitated spontaneous $Ca^{2+}$ fluctuations (*Figure 1A*). In particular, fluctuation-positive cell ratio and fluctuation amplitude were remarkably elevated in response to the CNP treatments. In contrast, ANP treatments exerted no effects on $Ca^{2+}$ fluctuations in growth plate chondrocytes.

In chondrocyte-specific *Npr2*-knockout mice (*Npr2*fl/fl, *Col2a1-Cre*+/−), Cre recombinase is expressed under the control of the collagen type 2α1 gene promoter and thus inactivates the floxed *Npr2* alleles in a chondrocyte-specific manner (*Nakao et al., 2015*). Our RT-PCR analysis indicated that the floxed *Npr2* gene was largely inactivated in the growth plates prepared from the E17.5 mutant embryos, but such recombination events were not detected in other tissues examined (*Figure 1—figure supplement 1A, B*). Accordingly, *Npr2* mRNA contents in the mutant growth plates were reduced to less than 40% of controls (*Figure 1—figure supplement 1C*), despite the growth plate prepara-tions contain not only chondrocytes but also perichondrium-resident cells including undifferentiated mesenchymal cells and immature chondroblasts. Further RT-PCR analysis detected similar expression

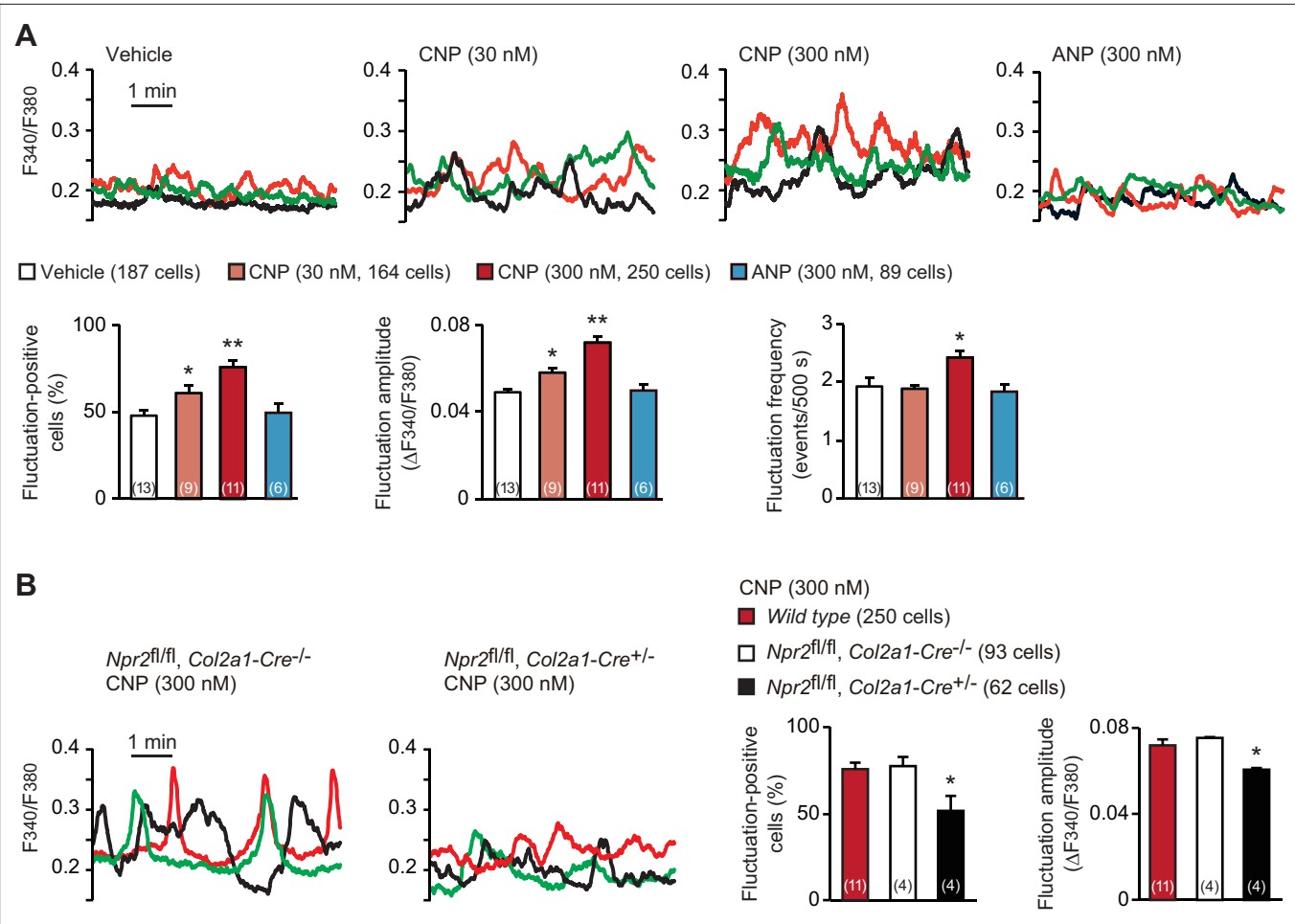

**Figure 1.** C-type natriuretic peptide (CNP)-induced facilitation of $Ca^{2+}$ fluctuations in growth plate chondrocytes. (**A**) Fura-2 imaging of round chondrocytes pretreated with or without natriuretic peptides. Femoral bone slices prepared from wild-type C57BL embryos were pretreated with or without CNP and atrial natriuretic peptide (ANP), and subjected to $Ca^{2+}$ imaging. Representative recording traces from three cells are shown in each pretreatment group (upper panels). The effects of CNP and ANP pretreatments on spontaneous $Ca^{2+}$ fluctuations are summarized (lower graphs). The fluctuation-positive cell ratio, fluctuation amplitude and frequency were statistically analyzed, and significant differences from the control vehicle pretreatment are marked with asterisks (*$p < 0.05$ and **$p < 0.01$ in one-way analysis of variance (ANOVA) and Dunnett's test). The data are presented as the means ± standard error of the mean (SEM). with *n* values indicating the number of examined mice. (**B**) Fura-2 imaging of round chondrocytes prepared from chondrocyte-specific *Npr2*-knockout (*Npr2*fl/fl, *Col2a1-Cre*+/−) and control (*Npr2*fl/fl, *Col2a1-Cre*−/−) mice. The bone slices were pretreated with CNP, and then subjected to $Ca^{2+}$ imaging. Representative recording traces are shown (left panels) and the CNP-pretreated effects are summarized (right graphs); significant differences from the wild-type group are marked with asterisks (*$p < 0.05$ in one-way ANOVA and Tukey's test). The data are presented as the means ± SEM with *n* values indicating the number of examined mice.

The online version of this article includes the following source data and figure supplement(s) for figure 1:

**Source data 1.** Related to *Figure 1A*.

**Source data 2.** Related to *Figure 1B*.

**Figure supplement 1.** Chondrocyte-specific *Npr2 ablation*.

**Figure supplement 1—source data 1.** Related to *Figure 1—figure supplement 1B*.

**Figure supplement 1—source data 2.** Related to *Figure 1—figure supplement 1B*.

**Figure supplement 1—source data 3.** Related to *Figure 1—figure supplement 1C*.

**Figure supplement 1—source data 4.** Related to *Figure 1—figure supplement 1D*.

**Figure supplement 2.** Gene expression analysis in wild-type growth plate chondrocytes.

**Figure supplement 2—source data 1.** Related to *Figure 1—figure supplement 2*.

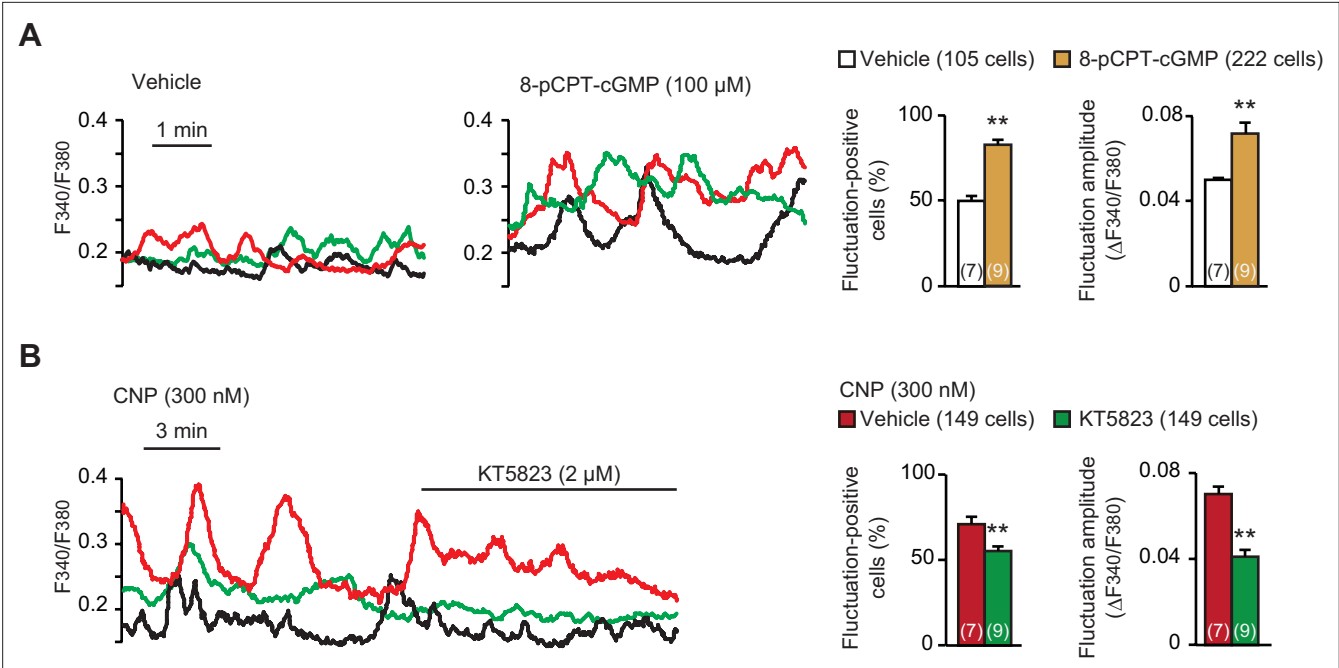

**Figure 2.** Contribution of cGMP-dependent protein kinase (PKG) to C-type natriuretic peptide (CNP)-facilitated Ca²⁺ fluctuations. (**A**) Facilitated Ca²⁺ fluctuations in round chondrocytes pretreated with the PKG activator 8-pCPT-cGMP. Wild-type bone slices were pretreated with or without the cGMP analog, and then subjected to Ca²⁺ imaging. Representative recording traces are shown (left panels), and the pharmacological effects are summarized (right graphs). Significant differences between control and 8-pCPT-cGMP pretreatments are marked with asterisks (**p < 0.01 in t-test). The data are presented as the means ± standard error of the mean (SEM) with *n* values indicating the number of examined mice. (**B**) Attenuation of CNP-facilitated Ca²⁺ fluctuations by the PKG inhibitor KT5823. Wild-type bone slices were pretreated with CNP, and then subjected to Ca²⁺ imaging. Representative recording traces are shown (left panel), and KT5823-induced effects are summarized (right graphs). Significant KT5823-induced shifts are marked with asterisks (**p < 0.01 in t-test). The data are presented as the means ± SEM with *n* values indicating the number of examined mice.

The online version of this article includes the following source data for figure 2:

**Source data 1.** Related to *Figure 2A*.

**Source data 2.** Related to *Figure 2B*.

of representative chondrogenic marker genes between chondrocyte-specific *Npr2*-knockout and control bones, suggesting that the *Npr2* deficiency does not affect fundamental chondrogenesis in growth plates (*Figure 1—figure supplement 1D*). In contrast to the imaging observations in wild-type and control bone slices, CNP treatments failed to enhance Ca²⁺ fluctuations in the mutant round chondrocytes prepared from the chondrocyte-specific *Npr2*-knockout mice (*Figure 1B*). Therefore, CNP seems to facilitate spontaneous Ca²⁺ fluctuations downstream of NPR2 activation in growth plate chondrocytes.

## Activated PKG facilitates spontaneous Ca²⁺ fluctuations

CNP binds to NPR2 to activate its intrinsic guanylate cyclase and thus stimulates PKG by elevating cellular cGMP contents (*Nakao et al., 1996*). CNP also binds to NPR3 which acts as a decoy receptor for ligand clearance, but the *Npr3* gene seemed to be inactive in growth plate chondrocytes (*Figure 1—figure supplement 2*). Next, we pharmacologically verified the contribution of PKG to CNP-facilitated Ca²⁺ fluctuations. The cGMP analog 8-(4-chlorophenylthio)-cyclic GMP (8-pCPT-cGMP) is widely used as a PKG-selective activator, while KT5823 is a typical PKG inhibitor. In wild-type growth plate chondrocytes pretreated with 8-pCPT-cGMP (100 µM for 1 hr), spontaneous Ca²⁺ fluctuations were remarkably facilitated (*Figure 2A*); both fluctuation-positive cell rate and fluctuation amplitude were highly increased. In contrast, the bath application of KT5823 (2 µM) clearly attenuated CNP-facilitated Ca²⁺ fluctuations within a short time frame (*Figure 2B*). Therefore, PKG activation seems to be essential for CNP-facilitated Ca²⁺ fluctuations in growth plate chondrocytes.

## Activated BK channels contribute to CNP-facilitated Ca²⁺ fluctuations

Spontaneous Ca²⁺ fluctuations are facilitated by activated BK channels in growth plate chondrocytes (*Qian et al., 2019*). Previous studies have established a functional link between PKG and BK channels in several cell types including smooth muscle and endothelial cells; activated PKG enhances BK channel gating by directly phosphorylating the α subunit KCNMA1 protein (*Dong et al., 2008*; *Fukao et al., 1999*; *White et al., 2000*). We thus examined whether altered BK channel activity is associated with CNP-facilitated Ca²⁺ fluctuations. The BK channel inhibitor paxilline (10 µM) exerted no obvious effects on basal Ca²⁺ fluctuations in nontreated chondrocytes. However, the same paxilline treatments remarkably inhibited CNP-facilitated Ca²⁺ fluctuations (*Figure 3A*); both fluctuation-positive cell ratio and fluctuation amplitude were clearly decreased after paxilline application. On the other hand, the BK channel activator NS1619 (30 µM) stimulated basal Ca²⁺ fluctuations in the growth plate chondrocytes prepared from control mice. The NS1619-induced effects were preserved in the mutant chondrocytes prepared from chondrocyte-specific *Npr2*-knockout mice (*Figure 3B*). Therefore, BK channel activation is likely involved in CNP-facilitated Ca²⁺ fluctuations in growth plate chondrocytes.

## Phospholipase C seems unrelated to CNP-facilitated Ca²⁺ fluctuations

Ca²⁺ fluctuations are maintained by phosphatidylinositol (PI) turnover in growth plate chondrocytes (*Qian et al., 2019*). Although it has been reported that activated PKG inhibits phospholipase C (PLC) in smooth muscle (*Guo et al., 2018*; *Huang et al., 2007*; *Nalli et al., 2014*; *Xia et al., 2001*), it might be possible that NPR2 activation enhances basal PLC activity to facilitate Ca²⁺ fluctuations. The PLC inhibitor U73122 (10 µM) remarkably inhibited basal Ca²⁺ fluctuations in nontreated chondrocytes: the fluctuation-positive cell ratio and fluctuation amplitude reduced less than half in response to U73122 application (*Figure 3—figure supplement 1*). U73122 was also effective for CNP-facilitated Ca²⁺ fluctuations, but the inhibitory efficiency seemed relatively weak compared to those on basal fluctuations. Given the different inhibitory effects, it is rather unlikely that PLC activation accompanies CNP-facilitated Ca²⁺ fluctuations.

PKG stimulates sarco/endoplasmic reticulum Ca²⁺-ATPase (SERCA) by phosphorylating the Ca²⁺ pump regulatory peptide phospholamban (PLN) in smooth and cardiac muscle cells (*Bibli et al., 2015*; *Raeymaekers et al., 1988*; *Lalli et al., 1999*), and activated Ca²⁺ pumps generally elevate stored Ca²⁺ contents and thus stimulate store Ca²⁺ release. RT-PCR data suggested that the *Pln* gene and the *Atp2a2* gene encoding SERCA2 are weakly active in growth plate chondrocytes (*Figure 1—figure supplement 2*). To examine the effects of CNP treatments on Ca²⁺ stores, we examined Ca²⁺ responses to the activation of Gq-coupled lysophosphatidic acid (LPA) receptors (*Figure 3—figure supplement 2A*) and the Ca²⁺ pump inhibitor thapsigargin (*Figure 3—figure supplement 2B*). CNP- and vehicle-pretreated chondrocytes exhibited similar LPA-induced Ca²⁺ release and thapsigargin-induced Ca²⁺ leak responses. Therefore, CNP treatments seem ineffective for store Ca²⁺ pumps in growth plate chondrocytes. Moreover, the dose dependency of Ca²⁺ release by LPA (1–10 µM) was not altered between CNP- and vehicle-pretreated chondrocytes, implying that CNP does not affect basal PLC activity.

Among diverse Ca²⁺ handling-related proteins, PLC, PLN, and BK channels have been reported as PKG substrates, however, our observations suggested that both PLC and PLN receive no obvious functional regulation in CNP-treated chondrocytes. On the other hand, the paxilline treatments diminished CNP-facilitated Ca²⁺ fluctuations down to nontreated basal levels (*Figure 3A*), suggesting that activated BK channels predominantly contribute to CNP-facilitated Ca²⁺ fluctuations in growth plate chondrocytes.

## CNP induces BK channel-mediated hyperpolarization

To confirm the contribution of activated BK channels to CNP-facilitated Ca²⁺ fluctuations, we conducted confocal imaging using the voltage-dependent dye oxonol VI. In this imaging analysis, depolarization results in the accumulation of the dye into cells, in which the fractional fluorescence intensity, normalized to the maximum intensity monitored in the bath solution containing 100 mM KCl, is thus increased (*Figure 4A*, left panel). The fractional intensity of CNP-pretreated cells was significantly lower than that of nontreated cells in a normal bath solution (*Figure 4A*, middle graph), although both cells exhibited similar intensity shifts in high K⁺ bath solutions. Based on the recording data, we prepared a calibration plot for the relationship between the fractional intensity and theoretical

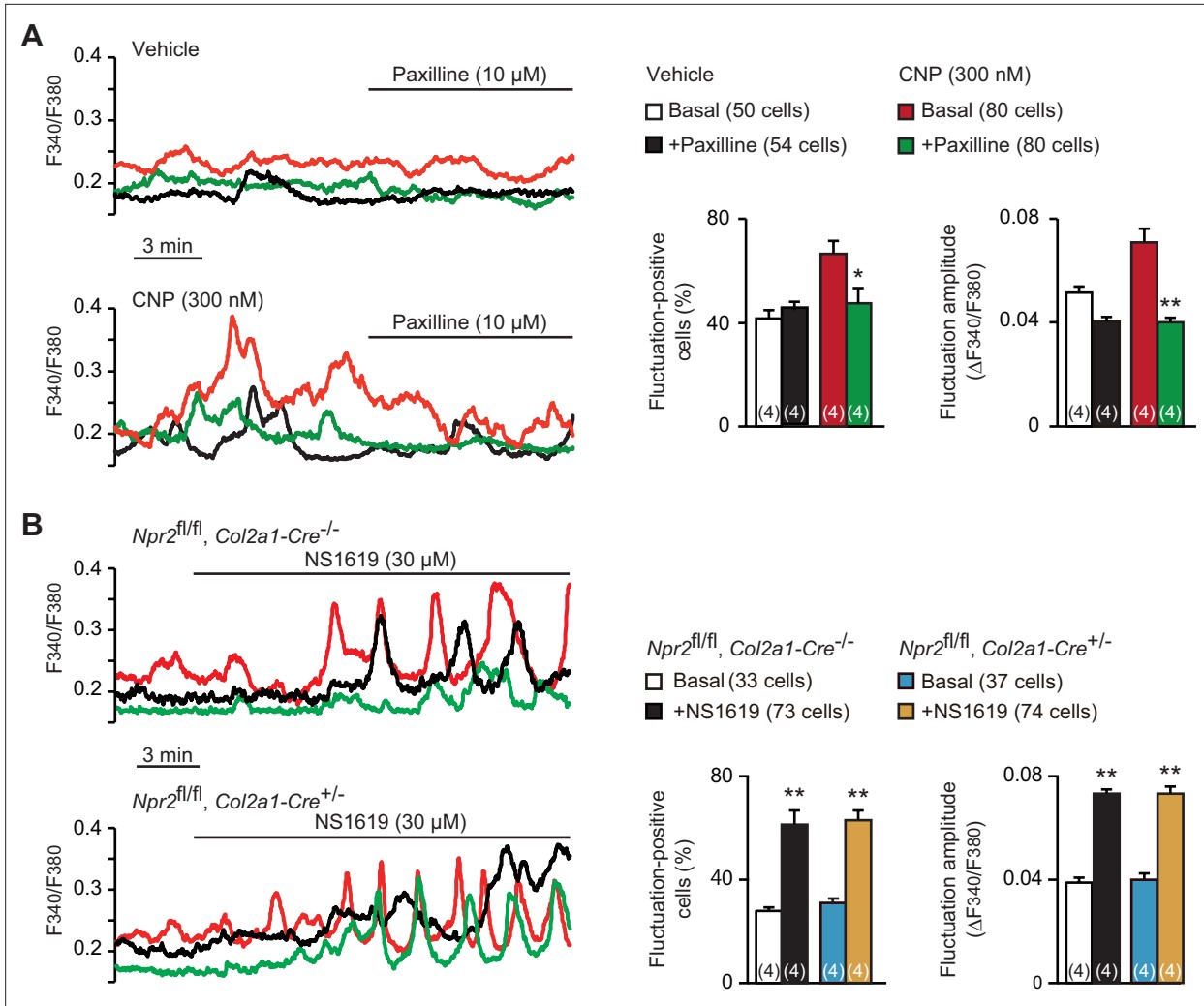

**Figure 3.** Contribution of BK channels to C-type natriuretic peptide (CNP)-facilitated $Ca^{2+}$ fluctuations. (**A**) Attenuation of CNP-facilitated $Ca^{2+}$ fluctuations by the BK channel inhibitor paxilline in round chondrocytes. Wild-type bone slices were pretreated with or without CNP, and then subjected to $Ca^{2+}$ imaging. Representative recording traces are shown (left panels), and paxilline-induced effects are summarized (right graphs). Significant paxilline-induced shifts are marked with asterisks (*p < 0.05 and **p < 0.01 in one-way analysis of variance (ANOVA) and Tukey's test). The data are presented as the means ± standard error of the mean (SEM) with *n* values indicating the number of examined mice. (**B**) $Ca^{2+}$ fluctuations facilitated by the BK channel activator NS1619 in *Npr2*-deficient chondrocytes. Bone slices were prepared from the chondrocyte-specific *Npr2*-knockout and control embryos, and NS1619-induced effects were examined in $Ca^{2+}$ imaging. Representative recording traces are shown (left panels), and the effects of NS1619 are summarized (right graphs). Significant NS1619-induced shifts are marked with asterisks (**p < 0.01 in one-way ANOVA and Tukey's test). The data are presented as the means ± SEM with *n* values indicating the number of examined mice.

The online version of this article includes the following source data and figure supplement(s) for figure 3:

**Source data 1.** Related to *Figure 3A*.

**Source data 2.** Related to *Figure 3B*.

**Figure supplement 1.** Effects of phospholipase C (PLC) inhibitor U73122 on C-type natriuretic peptide (CNP)-facilitated $Ca^{2+}$ fluctuations.

**Figure supplement 1—source data 1.** Related to *Figure 3—figure supplement 1*.

**Figure supplement 2.** Store $Ca^{2+}$ release in C-type natriuretic peptide (CNP)-treated round chondrocytes.

**Figure supplement 2—source data 1.** Related to *Figure 3—figure supplement 2A*.

**Figure supplement 2—source data 2.** Related to *Figure 3—figure supplement 2B*.

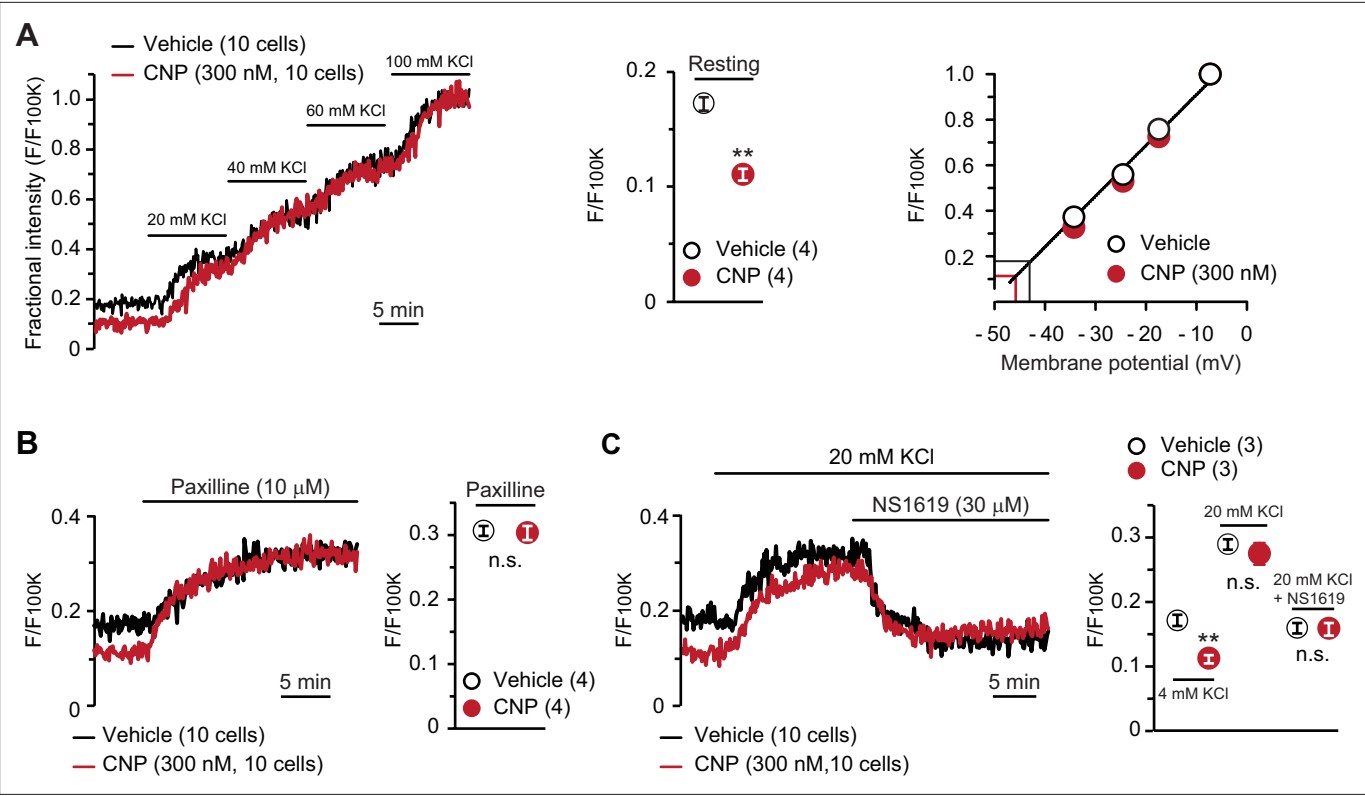

**Figure 4.** BK channel-mediated hyperpolarization induced by C-type natriuretic peptide (CNP). (**A**) Oxonol VI imaging of round chondrocytes pretreated with or without CNP. Wild-type bone slices were pretreated with or without CNP, and then subjected to membrane potential imaging. During contiguous treatments with high-K⁺ solutions, cellular fluorescence intensities were monitored and normalized to the maximum value in the 100 mM KCl-containing solution to yield the fractional intensity (left panel). The resting fractional intensities were quantified and statistically analyzed in CNP- and vehicle-pretreated cells (middle graph). For preparing the calibration plot (right panel), the data from 10 cells in bathing solutions containing 4 (normal solution), 20, 40, 60, and 100 mM KCl are summarized; red and black lines indicate the estimated resting membrane potentials of CNP- and vehicle-pretreated cells, respectively. (**B**) Effects of the BK channel inhibitor paxilline on resting membrane potential in round chondrocytes. Recording data from 10 cells pretreated with or without CNP were averaged (left panel), and the fractional intensities elevated by paxilline are summarized (right graph). (**C**) Effects of the BK channel activator NS1619 on membrane potential in round chondrocytes. Recording data from 10 cells pretreated with or without CNP were averaged (left panel), and the fractional intensities in normal, 20 mM KCl and NS1619-containing 20 mM KCl solutions are summarized (right graph). Significant differences between CNP- and vehicle-pretreated cells are indicated by asterisks in (**A**) (**p < 0.01 in t-test) and in (**C**) (**p < 0.01 in one-way analysis of variance [ANOVA] and Dunn's test). The data are presented as the means ± standard error of the mean (SEM) with $n$ values indicating the number of examined mice.

The online version of this article includes the following source data for figure 4:

**Source data 1.** Related to *Figure 4A*.

**Source data 2.** Related to *Figure 4B*.

**Source data 3.** Related to *Figure 4C*.

membrane potential (*Figure 4A*, right panel). In the tentative linear correlation, resting potentials of −46.4 ± 0.2 and −43.6 ± 0.3 mV were estimated in CNP- and nontreated cells, respectively. The estimated potentials closely approximate the reported value from monitoring articular chondrocytes using sharp microelectrodes (*Clark et al., 2010*).

In pharmacological assessments, paxilline elevated fractional intensities to the same levels in CNP- and nontreated chondrocytes (*Figure 4B*). Moreover, NS1619 decreased fractional intensities to the same levels in both cells under 20 mM KCl bathing conditions, which enabled us to reliably evaluate the reducing intensity shifts (*Figure 4C*). The oxonol VI imaging data suggested that CNP treatments induce BK channel-mediated hyperpolarization and thus facilitate spontaneous Ca²⁺ fluctuations by enhancing Ca²⁺-driving forces in growth plate chondrocytes.

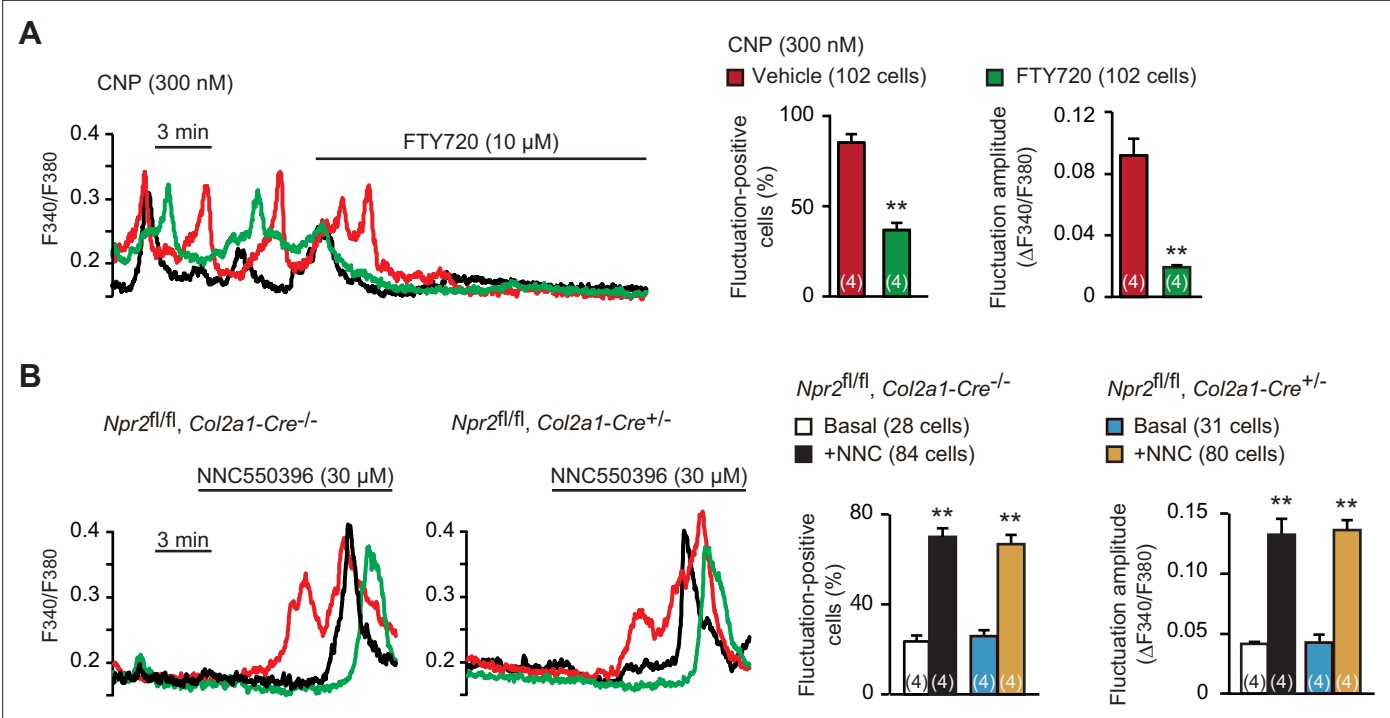

**Figure 5.** Enhanced TRPM7-mediated Ca$^{2+}$ entry by C-type natriuretic peptide (CNP) treatments. (**A**) Inhibition of CNP-facilitated Ca$^{2+}$ fluctuations by the TRPM7 inhibitor FTY720 in round chondrocytes. Wild-type bone slices were pretreated with CNP, and then subjected to Ca$^{2+}$ imaging. Representative recording traces are shown (left panel), and the effects of FTY720 are summarized (right graphs). Significant FTY720-induced shifts are marked with asterisks (**p < 0.01 in t-test). The data are presented as the means ± standard error of the mean (SEM) with *n* values indicating the number of examined mice. (**B**) Ca$^{2+}$ fluctuations facilitated by the TRPM7 channel activator NNC550396 in *Npr2*-deficient round chondrocytes. Bone slices were prepared from the chondrocyte-specific *Npr2*-knockout and control embryos, and NNC550396-induced effects were examined in Ca$^{2+}$ imaging. Representative recording traces are shown (left panels) and the effects of NNC550396 on Ca$^{2+}$ fluctuations are summarized (right graphs). Significant NNC550396-induced shifts in each genotype are marked with asterisks (**p < 0.01 in one-way analysis of variance [ANOVA] and Tukey's test). The data are presented as the means ± SEM with *n* values indicating the number of examined mice.

The online version of this article includes the following source data for figure 5:

**Source data 1.** Related to *Figure 5A*.

**Source data 2.** Related to *Figure 5B*.

## CNP enhances TRPM7-mediated Ca$^{2+}$ entry and CaMKII activity

Spontaneous Ca$^{2+}$ fluctuations are predominantly attributed to the intermissive gating of cell-surface TRPM7 channels in growth plate chondrocytes (*Qian et al., 2019*). For pharmacological characterization of TRPM7 channels, FTY720 is used as a typical inhibitor, while NNC550396 is an activator. As reasonably expected, bath application of FTY720 (10 μM) clearly diminished CNP-facilitated Ca$^{2+}$ fluctuations (*Figure 5A*). On the other hand, NNC550396 (30 μM) remarkably facilitated Ca$^{2+}$ fluctuations in nontreated chondrocytes, and this facilitation was preserved in the mutant chondrocytes prepared from chondrocyte-specific *Npr2*-knockout mice (*Figure 5B*). Therefore, CNP treatments likely facilitate TRPM7-mediated Ca$^{2+}$ influx in growth plate chondrocytes.

TRPM7-mediated Ca$^{2+}$ entry activates CaMKII in growth plate chondrocytes toward bone outgrowth (*Qian et al., 2019*), and cellular CaMKII activity can be estimated by immunochemically quantifying its autophosphorylated form. In immunocytochemical analysis, CNP-pretreated growth plate chondrocytes were more decorated with the antibody against phospho-CaMKII than nontreated control cells (*Figure 6A*). This CNP-facilitated decoration was abolished by the cotreatment of the CaMKII inhibitor KN93 (30 μM). This observation was further confirmed by Western blot analysis; CNP treatments increased the phospho-CaMKII population without affecting total CaMKII content in the cell lysates prepared from growth plates (*Figure 6B*). Therefore, CaMKII is likely activated downstream of enhanced TRPM7-mediated Ca$^{2+}$ entry in CNP-treated growth plate chondrocytes.

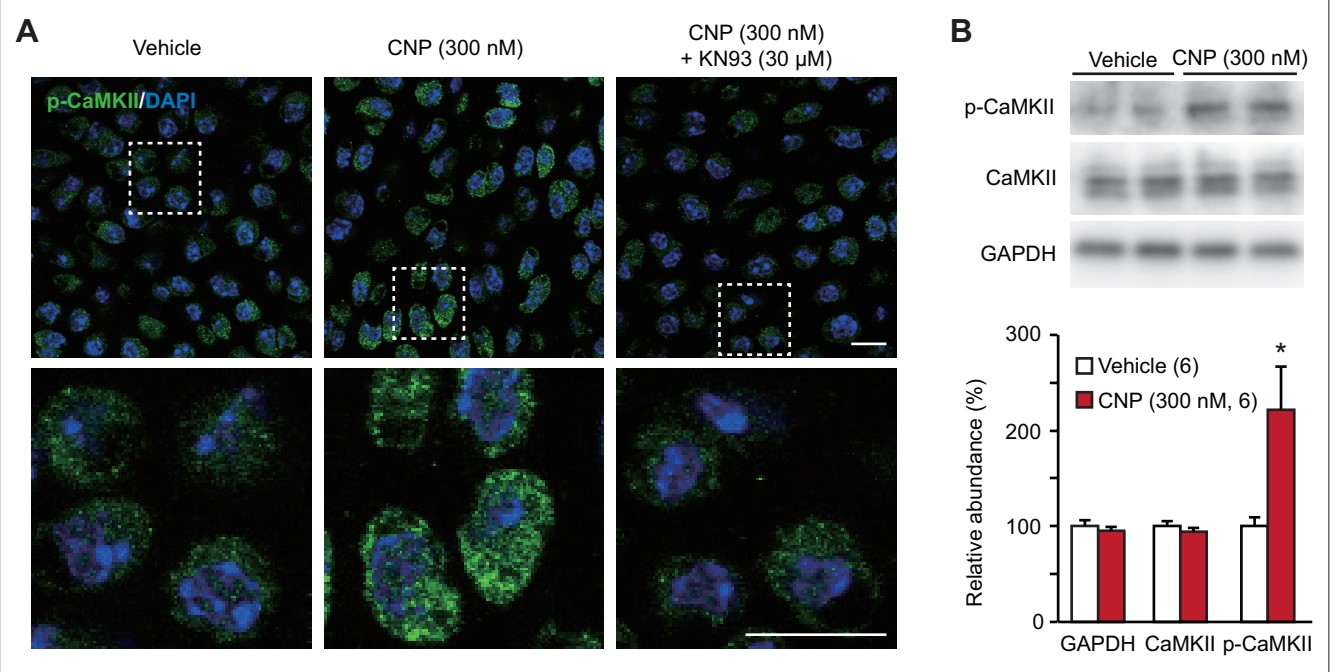

**Figure 6.** CaMKII activation in C-type natriuretic peptide (CNP)-treated round chondrocytes. (**A**) Immunohistochemical staining against phospho-CaMKII (p-CaMKII) in round chondrocytes. Wild-type bone slices were pretreated with or without CNP and the CaMKII inhibitor KN93, and then subjected to immunostaining with antibody to p-CaMKII. DAPI (4',6-diamidino-2-phenylindole) was used for nuclear staining. Lower panels show high-magnification views of white-dotted regions in upper panels (scale bars, 10 μm). (**B**) Immunoblot analysis of total CaMKII and p-CaMKII in growth plate cartilage. Growth plate lysates were prepared from wild-type bone slices pretreated with or without CNP, and subjected to immunoblot analysis with antibodies against total CaMKII and p-CaMKII (upper panel). Glyceraldehyde-3-phosphate dehydrogenase (GAPDH) was also analyzed as a loading control. The immunoreactivities observed were densitometrically quantified and are summarized (lower graph). A significant difference between CNP- and vehicle pretreatments is marked with an asterisk (*$p < 0.05$ in one-way analysis of variance [ANOVA] and Tukey's test). The data are presented as the means ± standard error of the mean (SEM) with $n$ values indicating the number of examined mice.

The online version of this article includes the following source data for figure 6:

**Source data 1.** Related to *Figure 6B*.

**Source data 2.** Related to *Figure 6B*.

## Pharmacologically activated BK channels facilitate bone outgrowth

Based on the present data from in vitro experiments, the novel CNP signaling route, represented as the NPR2-PKG-BK channel–TRPM7 channel–CaMKII axis, can be proposed in growth plate chondrocytes. We attempted to examine the proposed signaling axis in metatarsal bone culture, a widely used ex vivo model system for analyzing bone growth and endochondral ossification (*Houston et al., 2016*). CNP treatments expanded columnar chondrocytic zones without affecting cell densities to extend cultured wild-type metatarsal bones (*Figure 7—figure supplement 1*). The extension seemed to be mainly caused by enlarged extracellular matrix area, although CNP significantly dilated columnar cell sizes. In chondrocyte-specific *Trpm7*-knockout mice (*Trpm7*[fl/fl], *Col11a2*-Cre[+/−]), Cre recombinase is expressed under the control of the collagen type XI gene enhancer and promoter, and thus inactivates the floxed *Trpm7* alleles in cartilage cells (*Qian et al., 2019*). The bone rudiments prepared from control embryos (*Trpm7*[fl/fl], *Col11a2*-Cre[−/−]) regularly elongated during ex vivo culture, and their outgrowth was significantly stimulated by the supplementation with CNP (30 nM) into the culture medium (*Figure 7A*). The histological observation of the growth plate regions demonstrated that the CNP treatment extended the columnar chondrocyte zone. In contrast, the mutant rudiments prepared from the chondrocyte-specific *Trpm7*-knockout embryos were reduced in initial size and did not respond to the CNP supplementation (*Figure 7B*). Therefore, CNP-facilitated bone outgrowth seems to require TRPM7 channels expressed in growth plate chondrocytes.

In our proposed signaling axis, activated BK channels exert an essential role by converting the chemical signal into the electrical signal. We finally examined the effect of the BK channel activator

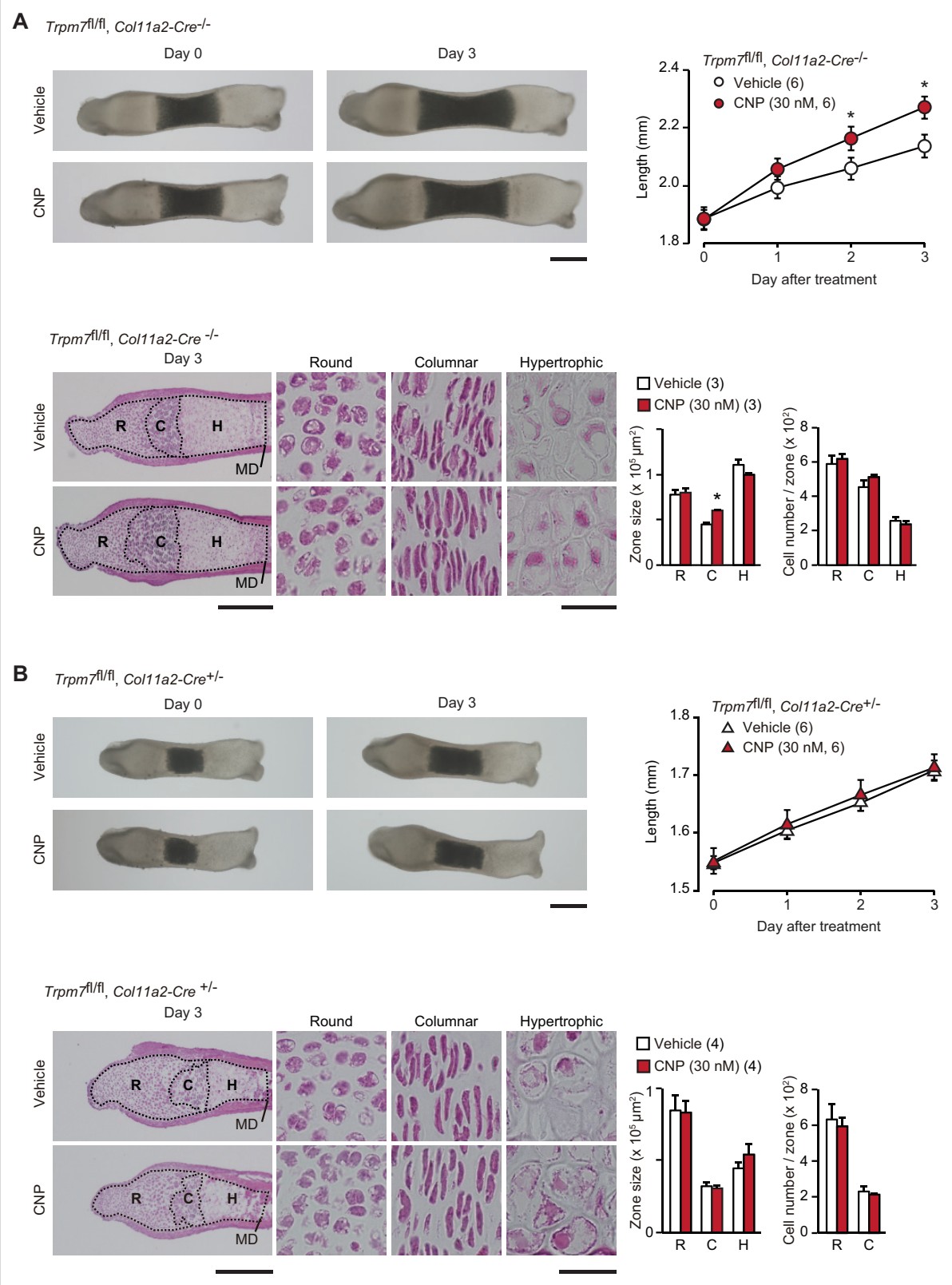

**Figure 7.** Contribution of TRPM7 channel to C-type natriuretic peptide (CNP)-facilitated bone outgrowth. Loss of CNP-facilitated outgrowth in *Trpm7*-deficient bones. Metatarsal rudiments isolated from control (*Trpm7*<sup>fl/fl</sup>, *Col11a2-Cre*<sup>−/−</sup>) embryos (**A**) and chondrocyte-specific *Trpm7*-knockout (*Trpm7*<sup>fl/fl</sup>, *Col11a2-Cre*<sup>+/−</sup>) embryos (**B**) were precultured in normal medium for 6 days, and then cultured in medium supplemented with or without CNP for 3 days. Representative images of cultured metatarsals are shown (upper left panels; scale bar, 0.3 mm), and longitudinal bone outgrowth during the

*Figure 7 continued on next page*

*Figure 7 continued*

CNP-supplemented period was statistically analyzed in each genotype group (upper right graphs). Growth plate images in longitudinal sections of 3-day cultured bones with or without CNP treatments are presented in lower left panels (scale bar, 0.3 mm), and their high-magnification views in the round (R), columnar (C), and hypertrophic (H) chondrocyte zones are shown in lower right panels (scale bar, 30 µm). MD, mid-diaphysis. Summary of graphical representations of zonal sizes containing round, columnar, and hypertrophic chondrocytes and number of cells in each zone is shown in lower right graphs. Significant CNP-supplemented effects are marked with asterisks (*p < 0.05 in *t*-test). The data are presented as the means ± standard error of the mean (SEM) with *n* values indicating the number of examined mice.

The online version of this article includes the following source data and figure supplement(s) for figure 7:

**Source data 1.** Related to *Figure 7A*.

**Source data 2.** Related to *Figure 7B*.

**Figure supplement 1.** Histological analysis of metatarsal bones treated with C-type natriuretic peptide (CNP).

**Figure supplement 1—source data 1.** Related to *Figure 7—figure supplement 1*.

**Figure supplement 2.** Proposed C-type natriuretic peptide (CNP)-evoked signaling in growth plate chondrocytes.

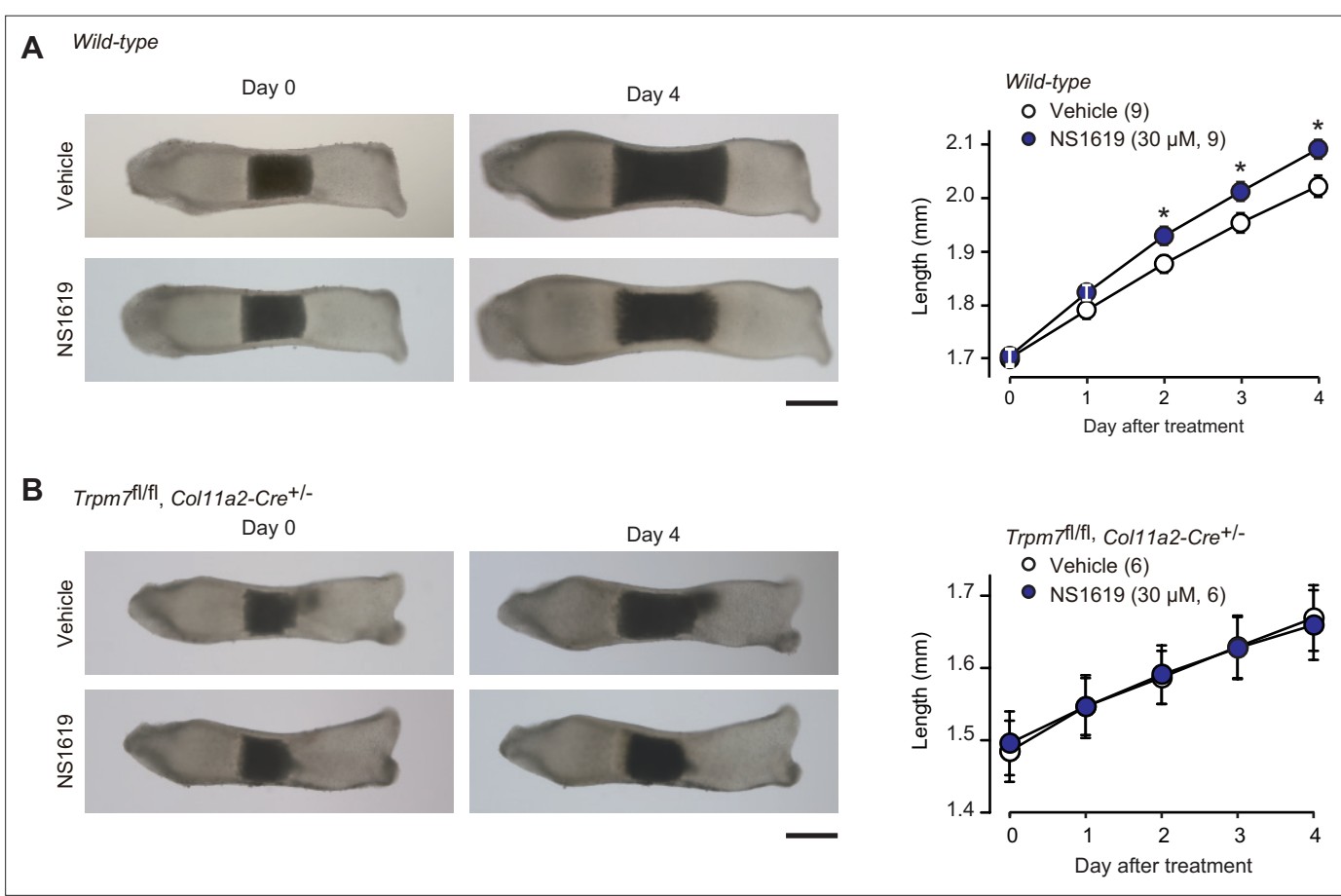

**Figure 8.** Facilitated bone outgrowth by BK channel activator. Stimulated bone outgrowth by the BK channel activator NS1619. Metatarsal rudiments isolated from wild-type (**A**) and the chondrocyte-specific *Trpm7*-knockout embryos (**B**) were precultured in normal medium for 5 days, and then cultured in medium supplemented with or without NS1619 for 4 days. Representative images of cultured metatarsals are shown (left panels; scale bar, 0.3 mm), and longitudinal bone outgrowth during the NS1619-supplemented period was statistically analyzed in each genotype group (right graphs). A significant NS1619-supplemented effect is marked with asterisks (*p < 0.05 in *t*-test). The data are presented as the means ± standard error of the mean (SEM) with *n* values indicating the number of examined mice.

The online version of this article includes the following source data for figure 8:

**Source data 1.** Related to *Figure 8A*.

**Source data 2.** Related to *Figure 8B*.

NS1619 on bone outgrowth (*Figure 8A*). NS1619 supplementation (30 µM) significantly stimulated the outgrowth of wild-type bone rudiments. In contrast, under the same culture conditions, no stimulation was detected in the mutant rudiments from the chondrocyte-specific *Trpm7*-deficient embryos (*Figure 8B*). The observations seem to support our conclusion that CNP activates BK channels and thus facilitates TRPM7-mediated Ca$^{2+}$ influx in growth plate chondrocytes for stimulating bone outgrowth.

## Discussion

We reported that in growth plate chondrocytes, PLC and BK channels maintain autonomic TRPM7-mediated Ca$^{2+}$ fluctuations, which potentiate chondrogenesis and bone growth by activating CaMKII (*Qian et al., 2019*). Based on the present data, together with the previous reports, we proposed a new CNP signaling axis in growth plate chondrocytes (*Figure 7—figure supplement 2A*). CNP-induced NPR2 activation elevates cellular cGMP content and thus activates PKG, leading to the phosphorylation of BK channels. The resulting BK channel activation induces cellular hyperpolarization to facilitate TRPM7-mediated Ca$^{2+}$ entry by enhancing the Ca$^{2+}$-driving force, leading to CaMKII activation. Therefore, it is likely that CaMKII activity is physiologically regulated by BK channels as a key player of the CNP signaling cascade. In a recent genetic study, several patients carrying loss-of-function mutations in the *KCNMA1* gene encoding BK channel α subunit were characterized by a novel syndromic growth deficiency associated with severe developmental delay, cardiac malformation, bone dysplasia, and dysmorphic features (*Liang et al., 2019*). In the *KCNMA1*-mutated disorder, CNP signaling likely fails to facilitate TRPM7-mediated Ca$^{2+}$ fluctuations in growth plate chondrocytes and resulting insufficient Ca$^{2+}$ entry may lead to systemic bone dysplasia associated with stunted growth plate cartilage. On the other hand, the origin of CNP may still be ambiguous in the signaling scheme. Transgenic mice overexpressing CNP in a chondrocyte-specific manner develop a prominent skeletal overgrowth phenotype, suggesting autocrine CNP signaling in developing bones (*Yasoda et al., 2004*). However, several genechip data in public databases indicate that prepro-CNP mRNA is abundantly expressed in the placenta among embryonic tissues (e.g., see the records under accession number GSE28277 in NCBI database). Therefore, it may be important to further examine which cell type primarily produces CNP to facilitate bone growth during embryonic development.

In our proposed CNP-signaling cascade, CaMKII is finally activated by TRPM7-mediated Ca$^{2+}$ influx in both round and columnar chondrocytes. However, it is still unclear how activated CaMKII contributes to bone outgrowth. Our obeservations in cultured metatarsal bones suggest that CNP expanded the columnar chondrocyte zone by stimulating the cell growth and enlarging the extracellular matrix area toward bone extension (*Figure 7*). The observations are roughly consistent with the previous studies using cultured tibias treated with CNP (*Yasoda et al., 1998*; *Miyazawa et al., 2002*). Therefore, activated CaMKII by TRPM7-mediated Ca$^{2+}$ influx probably phosphorylates key proteins controlling cell growth and extracellular matrix production in columner chondrocytes.

From a physiological point of view, it is interesting to note that the proposed CNP signaling axis has clear overlap with the nitric oxide (NO) and ANP/BNP signaling cascades for vascular relaxation (*Martel et al., 2010*; *Zois et al., 2014*; *Kubacka et al., 2018*). In blood vessels, NO is produced by endothelial cells in response to various stimuli including shear stress and acetylcholine, and activates soluble guanylate cyclase in neighboring vascular smooth muscle cells. ANP and BNP are released from the heart in response to pathological stresses, such as atrial distension and pressure overload, and are delivered to activate the receptor guanylate cyclase NPR1 in vascular muscle. In either case, the resulting cGMP elevation followed by PKG activation induces BK channel-mediated hyperpolarization and thus inhibits L-type Ca$^{2+}$ channel gating, leading to vascular dilation due to decreased Ca$^{2+}$ entry into vascular muscle. Therefore, activated BK channels inhibit the voltage-dependent Ca$^{2+}$ influx in vascular muscle cells regarded as excitable cells (*Figure 7—figure supplement 2B*). In contrast, activated BK channels reversely stimulate TRPM7-mediated Ca$^{2+}$ entry in growth plate chondrocytes classified as nonexcitable cells, because the channel activity is voltage independently maintained by the intrinsic PI turnover rate.

CNP is an effective therapeutic reagent for achondroplasia and divergent short statures (*Yasoda et al., 2004*; *Ueda et al., 2016*; *Yamashita et al., 2020*), and the phase III clinical trial of CNP therapy is completed successfully (*Nakao et al., 1996*). The proteins contributing to the CNP signaling axis may be new pharmaceutical targets for developing medications; in addition to NPR2, BK, and TRPM7

channels are reasonably considered promising targets. Moreover, phosphodiesterase subtypes might be useful targets, although the subtypes responsible for cGMP hydrolysis remain to be identified in growth plate chondrocytes. Chemical compounds specifically targeting the signaling axis defined in this study would be useful drugs for not only clinical treatment of developmental disorders but also artificially modifying body sizes in farm and pet animals.

# Materials and methods

## Key resources table

| Reagent type (species) or resource | Designation | Source or reference | Identifiers | Additional information |
|---|---|---|---|---|
| Strain, strain background (*Mus musculus*) | Mouse: C57BL/6J | The Jackson Laboratory | Jax: 000664 | |
| Strain, strain background (*Mus musculus*) | *Trpm7*fl/fl, *Col11a2-Cre* mice | **Qian et al., 2019** | N/A | |
| Strain, strain background (*Mus musculus*) | *Npr2*fl/fl, *Col2a1-Cre* mice | **Nakao et al., 2015** | N/A | |
| Antibody | Anti-phospho-CaMKII (Thr 286) (Rabbit monoclonal) | Cell Signaling Technology | Cat#12716; RRID: AB_2713889 | IF (1:200) WB (1:1000) |
| Antibody | Anti-CaMKII (Rabbit monoclonal) | Abcam | Cat#EP1829Y; RRID: AB_868641 | WB (1:1000) |
| Antibody | Anti-GAPDH (Rabbit polyclonal) | Sigma-Aldrich | Cat#G9545; RRID: AB_796208 | WB (1:10,000) |
| Antibody | Anti-rabbit IgG-HRP (Mouse monoclonal) | Santa Cruz | Cat#sc-2357; RRID: AB_628497 | 1:2000 |
| Antibody | Anti-rabbit Alexa Flour 488 (Goat polyclonal) | Invitrogen | Cat#A-11008; RRID: AB_143165 | 1:50 |
| Sequence-based reagent | Mouse *Npr1*_F | This paper | PCR primers | AACAAGGAGAACAGCAGCAAC |
| Sequence-based reagent | Mouse *Npr1*_R | This paper | PCR primers | TATCAAATGCCTCAGCCTGGA |
| Sequence-based reagent | Mouse *Npr2*_F | This paper | PCR primers | GGCCCCATCCCTGATGAAC |
| Sequence-based reagent | Mouse *Npr2*_R | This paper | PCR primers | CCTGGTACCCCCTTCCTGTA |
| Sequence-based reagent | Mouse *Npr3*_F | This paper | PCR primers | GGTATGGGGACTTCTCTGTG |
| Sequence-based reagent | Mouse *Npr3*_R | This paper | PCR primers | TCTGGTCTCATCTAGTCTCA |
| Sequence-based reagent | FlFor | This paper | PCR primers | GTAACCTGGGTAGACTAGTTGTTGG |
| Sequence-based reagent | DelFor | This paper | PCR primers | TGTTATTTTGTGAGATGACG |
| Sequence-based reagent | Rev | This paper | PCR primers | ATGGTGGAGGAGGTCTTTAATTCC |
| Sequence-based reagent | *Col2a1*-Cre_F | This paper | PCR primers | CGTTGTGAGTTGGATAGTTG |
| Sequence-based reagent | *Col2a1*-Cre_R | This paper | PCR primers | CATTGCTGTCACTTGGTCGT |
| Sequence-based reagent | Mouse *Prkg1*_F | This paper | PCR primers | ATGGACTTTTTGTGGGACTC |
| Sequence-based reagent | Mouse *Prkg1*_R | This paper | PCR primers | GGTTTTCATTGGATCTGGGC |
| Sequence-based reagent | Mouse *Prkg2*_F | This paper | PCR primers | TTGCGGAAGAAAATGATGTCG |
| Sequence-based reagent | Mouse *Prkg2*_R | This paper | PCR primers | GAATGGGGAGGTTGAGGAGAA |

*Continued on next page*

*Continued*

| Reagent type (species) or resource | Designation | Source or reference | Identifiers | Additional information |
|---|---|---|---|---|
| Sequence-based reagent | Mouse *Kcnma*_F | *Liu et al., 2021* | PCR primers | AATGCACTTCGAGGAGGCTA |
| Sequence-based reagent | Mouse *Kcnma*_R | *Liu et al., 2021* | PCR primers | CTCAGCCGGTAAATTCCAAA |
| Sequence-based reagent | Mouse *Kcnmb1*_F | This paper | PCR primers | ACAACTGTGCTGCCCCTCTA |
| Sequence-based reagent | Mouse *Kcnmb1*_R | This paper | PCR primers | CACTGTTGGTTTTGATCCCG |
| Sequence-based reagent | Mouse *Kcnmb2*_F | This paper | PCR primers | TCAGGAGACACCAACACTTC |
| Sequence-based reagent | Mouse *Kcnmb2*_R | This paper | PCR primers | AGTTAGTTTCACCATAGCAA |
| Sequence-based reagent | Mouse *Kcnmb3*_F | This paper | PCR primers | GTGGATGACGGGCTGGACTT |
| Sequence-based reagent | Mouse *Kcnmb3*_R | This paper | PCR primers | GCACTTGGGGTTGGTCCTGA |
| Sequence-based reagent | Mouse *Kcnmb4*_F | This paper | PCR primers | CTCCTGACCAACCCCAAGT |
| Sequence-based reagent | Mouse *Kcnmb4*_R | This paper | PCR primers | TAAAATAGCAAGTGAATGGC |
| Sequence-based reagent | Mouse *Kcnn1*_F | *Liu et al., 2021* | PCR primers | TCAAAAATGCTGCTGCAAAC |
| Sequence-based reagent | Mouse *Kcnn1*_R | *Liu et al., 2021* | PCR primers | TCGTTCACCTTCCCTTGTTC |
| Sequence-based reagent | Mouse *Kcnn2*_F | *Liu et al., 2021* | PCR primers | GATCTGGCAAAGACCCAGAA |
| Sequence-based reagent | Mouse *Kcnn2*_R | *Liu et al., 2021* | PCR primers | GAAGTCCCTTTGCTGCTGTC |
| Sequence-based reagent | Mouse *Kcnn3*_F | *Liu et al., 2021* | PCR primers | ACTTCAACACCCGATTCGTC |
| Sequence-based reagent | Mouse *Kcnn3*_R | *Liu et al., 2021* | PCR primers | GGAAAGGAACGTGATGGAGA |
| Sequence-based reagent | Mouse *Kcnn4*_F | *Liu et al., 2021* | PCR primers | GGCACCTCACAGACACACTG |
| Sequence-based reagent | Mouse *Kcnn4*_R | *Liu et al., 2021* | PCR primers | TTTCTCCGCCTTGTTGAACT |
| Sequence-based reagent | Mouse *Plcb1*_F | *Yamazaki et al., 2011* | PCR primers | CCCAAGTTGCGTGAACTTCT |
| Sequence-based reagent | Mouse *Plcb1*_R | *Yamazaki et al., 2011* | PCR primers | GTTGCCAAGCTGAAAACCTC |
| Sequence-based reagent | Mouse *Plcb2*_F | *Yamazaki et al., 2011* | PCR primers | ACATCCAGGAAGTGGTCCAG |
| Sequence-based reagent | Mouse *Plcb2*_R | *Yamazaki et al., 2011* | PCR primers | CGCACCGACTCCTTTACTTC |
| Sequence-based reagent | Mouse *Plcb3*_F | *Yamazaki et al., 2011* | PCR primers | CAGGCCAGCACAGAGACATA |
| Sequence-based reagent | Mouse *Plcb3*_R | *Yamazaki et al., 2011* | PCR primers | AGGATGCTGGCAATCAAATC |
| Sequence-based reagent | Mouse *Plcg1*_F | This paper | PCR primers | AACGCTTTGAGGACTGGAGA |

*Continued on next page*

*Continued*

| Reagent type (species) or resource | Designation | Source or reference | Identifiers | Additional information |
|---|---|---|---|---|
| Sequence-based reagent | Mouse *Plcg1*_R | This paper | PCR primers | CTCCTCAATCTCTCGCAAGG |
| Sequence-based reagent | Mouse *Plcg2*_F | This paper | PCR primers | AACCCCAACCCACACGAGTC |
| Sequence-based reagent | Mouse *Plcg2*_R | This paper | PCR primers | AATGTTTCACCTTGCCCCTG |
| Sequence-based reagent | Mouse *Trpm7*_F | *Qian et al., 2019* | PCR primers | ATTGCTTAGTTTTGGTGTTC |
| Sequence-based reagent | Mouse *Trpm7*_R | *Qian et al., 2019* | PCR primers | GATTGTCGGGAGAGTGGAGT |
| Sequence-based reagent | Mouse *Camk2a*_F | This paper | PCR primers | CACCACCATTGAGGACGAAG |
| Sequence-based reagent | Mouse *Camk2a*_R | This paper | PCR primers | GGTTCAAAGGCTGTCATTCC |
| Sequence-based reagent | Mouse *Camk2b*_F | This paper | PCR primers | AAGCAGATGGAGTCAAGCC |
| Sequence-based reagent | Mouse *Camk2b*_R | This paper | PCR primers | TGCTGTCGGAAGATTCCAGG |
| Sequence-based reagent | Mouse *Camk2d*_F | This paper | PCR primers | GATAAACAACAAAGCCAACG |
| Sequence-based reagent | Mouse *Camk2d*_R | This paper | PCR primers | GTAAGCCTCAAAGTCCCCAT |
| Sequence-based reagent | Mouse *Camk2g*_F | This paper | PCR primers | CAAGAACAGCAAGCCTATCC |
| Sequence-based reagent | Mouse *Camk2g*_R | This paper | PCR primers | CCTCTGACTGACTGGTGCGA |
| Sequence-based reagent | Mouse *Pde2a*_F | This paper | PCR primers | ATCTTTGACCACTTCTCTCG |
| Sequence-based reagent | Mouse *Pde2a*_R | This paper | PCR primers | CATAACCCACTTCAGCCATC |
| Sequence-based reagent | Mouse *Pde3a*_F | This paper | PCR primers | AACTATACCTGCTCGGACTC |
| Sequence-based reagent | Mouse *Pde3a*_R | This paper | PCR primers | TTCGTGCGGCTTTATGCTGG |
| Sequence-based reagent | Mouse *Pde3b*_F | This paper | PCR primers | ATTCCAAAGCAGAGGTCATC |
| Sequence-based reagent | Mouse *Pde3b*_R | This paper | PCR primers | GTTAGAGAGCCAGCAGACAC |
| Sequence-based reagent | Mouse *Pde5a*_F | This paper | PCR primers | GACCCTTGCGTTGCTCATTG |
| Sequence-based reagent | Mouse *Pde5a*_R | This paper | PCR primers | TGATGGAGTGACAGTACAGC |
| Sequence-based reagent | Mouse *Pde6a*_F | This paper | PCR primers | AACCCACCCGCTGACCACTG |
| Sequence-based reagent | Mouse *Pde6a*_R | This paper | PCR primers | CTCTTCCTTCTTGTTGACGA |
| Sequence-based reagent | Mouse *Pde6b*_F | This paper | PCR primers | TCCGGGCCTATCTAAACTGC |
| Sequence-based reagent | Mouse *Pde6b*_R | This paper | PCR primers | AGAAGACAATTTCCCGGCCAT |

*Continued on next page*

*Continued*

| Reagent type (species) or resource | Designation | Source or reference | Identifiers | Additional information |
|---|---|---|---|---|
| Sequence-based reagent | Mouse *Pde6c*_F | This paper | PCR primers | TTGCTCAGGAAATGGTTATG |
| Sequence-based reagent | Mouse *Pde6c*_R | This paper | PCR primers | GAAACAGAACTCGTACAGGT |
| Sequence-based reagent | Mouse *Pde6d*_F | This paper | PCR primers | CCCAAGAAAATCCTCAAGTG |
| Sequence-based reagent | Mouse *Pde6d*_R | This paper | PCR primers | ACAAAGCCAAACTCGAAGAA |
| Sequence-based reagent | Mouse *Pde6g*_F | This paper | PCR primers | AAGGGTGAGATTCGGTCAGC |
| Sequence-based reagent | Mouse *Pde6g*_R | This paper | PCR primers | TCATCCCCAAACCCTTGCAC |
| Sequence-based reagent | Mouse *Pde6h*_F | This paper | PCR primers | GGCAGACTCGACAGTTCAAGA |
| Sequence-based reagent | Mouse *Pde6h*_R | This paper | PCR primers | CTCCAGATGGCTGAACGCT |
| Sequence-based reagent | Mouse *Pde10a*_F | This paper | PCR primers | CATCCGCAAAGCCATCATCG |
| Sequence-based reagent | Mouse *Pde10a*_R | This paper | PCR primers | TCTCATCACCCTCAGCCCAG |
| Sequence-based reagent | Mouse *Lpar1*_F | This paper | PCR primers | GCTTGGTGCCTTTATTGTCT |
| Sequence-based reagent | Mouse *Lpar1*_R | This paper | PCR primers | GGTAGGAGTAGATGATGGGG |
| Sequence-based reagent | Mouse *Lpar2*_F | This paper | PCR primers | AGTGTGCTGGTATTGCTGAC |
| Sequence-based reagent | Mouse *Lpar2*_R | This paper | PCR primers | TTTGATGGAGAGCCTGGCAG |
| Sequence-based reagent | Mouse *Lpar3*_F | This paper | PCR primers | ACTTTCCCTTCTACTACCTG |
| Sequence-based reagent | Mouse *Lpar3*_R | This paper | PCR primers | GTCTTTCCACAGCAATAACC |
| Sequence-based reagent | Mouse *Lpar4*_F | This paper | PCR primers | CCTCAGTGGTGGTATTTCAG |
| Sequence-based reagent | Mouse *Lpar4*_R | This paper | PCR primers | CACAGAAGAACAAGAAACAT |
| Sequence-based reagent | Mouse *Lpar5*_F | This paper | PCR primers | AACACGACTTCTACCAACAG |
| Sequence-based reagent | Mouse *Lpar5*_R | This paper | PCR primers | AAGACCCAGAGAGCCAGAGC |
| Sequence-based reagent | Mouse *Lpar6*_F | This paper | PCR primers | TACTTTGCCATTTCGGATTT |
| Sequence-based reagent | Mouse *Lpar6*_R | This paper | PCR primers | GCACTTCCTCCCATCACTGT |
| Sequence-based reagent | Mouse *Atp2a1*_F | *Liu et al., 2021* | PCR primers | CAAAACAGGGACCCTCACCA |
| Sequence-based reagent | Mouse *Atp2a1*_R | *Liu et al., 2021* | PCR primers | GCCAGTGATGGAGAACTCGT |
| Sequence-based reagent | Mouse *Atp2a2*_F | *Liu et al., 2021* | PCR primers | AAACCAGATGTCCGTGTGCA |

*Continued on next page*

*Continued*

| Reagent type (species) or resource | Designation | Source or reference | Identifiers | Additional information |
|---|---|---|---|---|
| Sequence-based reagent | Mouse *Atp2a2*_R | *Liu et al., 2021* | PCR primers | TGATGGCACTTCACTGGCTT |
| Sequence-based reagent | Mouse *Atp2a3*_F | *Liu et al., 2021* | PCR primers | CCTCGGTCATCTGCTCTGAC |
| Sequence-based reagent | Mouse *Atp2a3*_R | *Liu et al., 2021* | PCR primers | CGTGGTACCCGAAATGGTGA |
| Sequence-based reagent | Mouse *Pln*_F | This paper | PCR primers | TACCTCACTCGCTCGGCTAT |
| Sequence-based reagent | Mouse *Pln*_R | This paper | PCR primers | TGACGGAGTGCTCGGCTTTA |
| Sequence-based reagent | Mouse *Sox9*_F | *Qian et al., 2019* | PCR primers | AGGAAGCTGGCAGACCAGTA |
| Sequence-based reagent | Mouse *Sox9*_R | *Qian et al., 2019* | PCR primers | CGTTCTTCACCGACTTCCTC |
| Sequence-based reagent | Mouse *Sox5*_F | *Qian et al., 2019* | PCR primers | CTCGCTGGAAAGCTATGACC |
| Sequence-based reagent | Mouse *Sox5*_R | *Qian et al., 2019* | PCR primers | GATGGGGATCTGTGCTTGTT |
| Sequence-based reagent | Mouse *Sox6*_F | *Qian et al., 2019* | PCR primers | GGATTGGGGAGTACAAGCAA |
| Sequence-based reagent | Mouse *Sox6*_R | *Qian et al., 2019* | PCR primers | CATCTGAGGTGATGGTGTGG |
| Sequence-based reagent | Mouse *Runx2*_F | *Qian et al., 2019* | PCR primers | GCCGGGAATGATGAGAACTA |
| Sequence-based reagent | Mouse *Runx2*_R | *Qian et al., 2019* | PCR primers | GGACCGTCCACTGTCACTTT |
| Sequence-based reagent | Mouse *Pthlh*_F | *Qian et al., 2019* | PCR primers | CTCCCAACACCAAAAACCAC |
| Sequence-based reagent | Mouse *Pthlh*_R | *Qian et al., 2019* | PCR primers | GCTTGCCTTTCTTCTTCTTC |
| Sequence-based reagent | Mouse *Acan*_F | *Qian et al., 2019* | PCR primers | CCTCACCATCCCCTGCTACT |
| Sequence-based reagent | Mouse *Acan*_R | *Qian et al., 2019* | PCR primers | ACTTGATTCTTGGGGTGAGG |
| Sequence-based reagent | Mouse *Col10a1*_F | *Qian et al., 2019* | PCR primers | CAAGCCAGGCTATGGAAGTC |
| Sequence-based reagent | Mouse *Col10a1*_R | *Qian et al., 2019* | PCR primers | AGCTGGGCCAATATCTCCTT |
| Sequence-based reagent | Mouse *Col2a1*_F | *Qian et al., 2019* | PCR primers | CACACTGGTAAGTGGGGCAAGACCG |
| Sequence-based reagent | Mouse *Col2a1*_R | *Qian et al., 2019* | PCR primers | GGATTGTGTTGTTTCAGGGTTCGGG |
| Sequence-based reagent | Mouse *18* S_F | *Qian et al., 2019* | PCR primers | AGACAAATCGCTCCACCAAC |
| Sequence-based reagent | Mouse *18* S_R | *Qian et al., 2019* | PCR primers | CTCAACACGGGAAACCTCAC |
| Sequence-based reagent | Mouse *Actb*_F | *Qian et al., 2019* | PCR primers | CATCCGTAAAGACCTCTATGCCAAC |
| Sequence-based reagent | Mouse *Actb*_R | *Qian et al., 2019* | PCR primers | ATGGAGCCACCGATCCACA |

*Continued on next page*

*Continued*

| Reagent type (species) or resource | Designation | Source or reference | Identifiers | Additional information |
|---|---|---|---|---|
| Sequence-based reagent | Mouse *Gapdh*_F | **Qian et al., 2019** | PCR primers | TGTGTCCGTCGTGGATCTGA |
| Sequence-based reagent | Mouse *Gapdh*_R | **Qian et al., 2019** | PCR primers | TTGCTGTTGAAGTCGCAGGAG |
| Peptide, recombinant protein | ANP (Human, 1–28) | Peptide Institute | Cat#4135 | |
| Peptide, recombinant protein | CNP-22 (Human) | Peptide Institute | Cat#4229 | |
| Commercial assay or kit | Amersham ECL Prime Western Blotting Detection Reagent | Cytiva | Cat#RPN2232 | |
| Commercial assay or kit | ISOGEN | NipponGene | Cat#319-90211 | |
| Commercial assay or kit | ReverTra Ace qPCR RT Master Mix with gDNA Remover | TOYOBO | Cat#FSQ-301 | |
| Chemical compound, drug | FTY720 | Sigma-Aldrich | SML0700; CAS: 162359-56-0 | |
| Chemical compound, drug | Fura-2 AM | DOJINDO | F025; CAS: 108964-32-5 | |
| Chemical compound, drug | Hyaluronidase from sheep testes | Sigma-Aldrich | H2126; CAS: 37326-33-3 | |
| Chemical compound, drug | KN93 | WAKO | 115-00641; CAS: 139298-40-1 | |
| Chemical compound, drug | KT5823 | Cayman Chemical | 10010965; CAS: 126643-37-6 | |
| Chemical compound, drug | NNC 550396 dihydrochloride | Tocris Bioscience | 2268; CAS: 357400-13-6 | |
| Chemical compound, drug | NS1619 | Sigma-Aldrich | N170; CAS: 153587-01-0 | |
| Chemical compound, drug | 1-Oleoyl lysophosphatidic acid | Cayman Chemical | 62215: CAS: 325465-93-8 | |
| Chemical compound, drug | Oxonol VI | Sigma-Aldrich | 75926; CAS: 64724-75-0 | |
| Chemical compound, drug | Paxilline | Tocris Bioscience | 2006; CAS: 57186-25-1 | |
| Chemical compound, drug | 8-pCPT-cGMP | Biolog | C009; CAS: 51239-26-0 | |
| Chemical compound, drug | Thapsigargin | Nacalai Tesque | 33637-31; CAS: 67526-95-8 | |
| Chemical compound, drug | U73122 | Sigma-Aldrich | U6756; CAS: 112648-68-7 | |
| Software, algorithm | Adobe Ilustrator | Adobe Systems | http://www.adobe.com/products/illustrator.html | |
| Software, algorithm | GraphPad Prism v7 | GraphPad | https://www.graphpad.com/ | |
| Software, algorithm | ImageJ | N/A | https://imagej.nih.gov/ij/ | |
| Software, algorithm | Leica Application Suite X | Leica MIcrosystems | https://www.leica-microsystems.com/products/microscope-software/p/leica-las-x-ls/ | |

## Reagents, primers, and mice

Reagents and antibodies used in this study, and synthetic primers used for RT-PCR analysis and mouse genotyping are listed in Key Resourses Table. C57BL mice were used as wild-type mice in this study. Chondrocyte-specific *Trpm7*-knockout mice with C57BL genetic background were generated by crossing *Trpm7*^fl/fl mice (**Qian et al., 2019**) with transgenic mice carrying *Col11a2-Cre*, originally

designated as *11Enh-Cre* (*Iwai et al., 2008*) Using primer sets for detecting *Col11a2-Cre* transgene and *Trpm7* alleles, we previously reported that *Trpm7* is specifically inactivated in cartilage tissues from the *Trpm7*flfl, *Col11a2-Cre*⁺/⁻ mice (*Qian et al., 2019*). Chondrocyte-specific *Npr2*-knockout mice with C57BL background were generated as previously described (*Nakao et al., 2015*), and we designed primers for detecting the *Col2a1-Cre* transgene and the floxed *Npr2* gene in this study (*Figure 1—figure supplement 1*).

## Bone slice preparations

Femoral bones were isolated from E17.5 mice and immersed in a physiological salt solution (PSS): (in mM) 150 NaCl, 4 KCl, 1 $MgCl_2$, 2 $CaCl_2$, 5.6 glucose, and 5 2-[4-(2-Hydroxyethyl)-1-piperazinyl]ethane-sulfonic acid (HEPES, pH 7.4). Longitudinal bone slices (~40 µm thickness) were prepared using a vibrating microslicer (DTK-1000N, Dosaka EM Co., Japan) as previously described (*Qian et al., 2019*).

## Ca²⁺ imaging

Fura-2 $Ca^{2+}$ imaging of bone slices was performed as previously described (*Qian et al., 2019*). Briefly, bone slices placed on glass-bottom dishes (Matsunami, Japan) were incubated in PSS containing 15 µM Fura-2AM for 1 hr at 37°C. Fluorescence microscopy distinguished round, columnar, and hypertrophic chondrocytes with characteristic morphological features in the bone slices loaded with Fura-2. For ratiometric imaging, excitation light of 340 and 380 nm was alternately delivered, and emission light of >510 nm was detected by a cooled EM-CCD camera (Model C9100-13; Hamamatsu Photonics, Japan) mounted on an upright fluorescence microscope (DM6 FS, Leica) using a ×40 water-immersion objective (HCX APO L, Leica). In typical measurements, ~30 round chondrocytes were randomly examined in each slice preparation to select the $Ca^{2+}$ fluctuation-positive cells generating spontaneous events (>0.025 in Fura-2 ratio) using commercial software (Leica Application Suite X), and recording traces from the positive cells were then analyzed using Fiji/ImageJ software (US. NIH) for examining $Ca^{2+}$ fluctuation amplitude and frequency. Imaging experiments were performed at room temperature (23–25°C) and PSS was used as the normal bathing solution. For the pretreatments of CNP, ANP, and 8-pCPT-cGMP, bone slices were immersed in PSS with the indicated compound for 1 hr at room temperature after Fura-2 loading.

## Membrane potential monitoring

Bone slices were perfused with the PSS containing 200 nM oxonol VI at room temperature and analyzed as previously described (*Yamazaki et al., 2011*). To prepare the calibration plot showing the relationship between the fluorescence intensity and membrane potential, saline solutions containing 20, 40, 60, or 100 mM KCl were used as bathing solutions. Fluorescence images with excitation at 559 nm and emission at >606 nm were captured at a sampling rate of ~7.0 s using a confocal laser scanning microscope (FV1000; Olympus).

## Immunochemical analysis of CaMKII

Bone slices were pretreated with or without CNP were subjected to immunochemical assessments as previously described (*Li et al., 2011*). Briefly, for immunohistochemical analysis, bone slices were fixed in 4% paraformaldehyde and treated with 1% hyaluronidase to enhance immunode-tection (*Ahrens and Dudley, 2011*; *Mouser et al., 2016*). After blocking with fetal bovine serum-containing solution, bone slices were reacted with primary and Alexa 488-conjugated secondary antibodies and observed with a confocal microscope (FV1000; Olympus). For immunoblot analysis, bone slices were lysed in the buffer containing 4% sodium deoxycholate, 20 mM Tris–HCl (pH 8.8) and a phosphatase inhibitor cocktail (100 mM NaF, 10 mM $Na_3PO_4$, 1 mM $Na_2VO_3$, and 20 mM β-glycerophosphate). The resulting lysate proteins were electrophoresed on sodium dodecyl sulfate–polyacrylamide gels and electroblotted onto nylon membranes for immunodetection using primary and Horseradish peroxidase (HRP)-conjugated secondary antibodies. Antigen proteins were visualized using a chemiluminescence reagent and image analyzer (Amersham Imager 600, Cytiva). The immunoreactivities yielded were quantitatively analyzed by means of Fiji/ImageJ software.

## Metatarsal organ culture

Metatarsal bone rudiments were cultured as previously described (*Houston et al., 2016*). Briefly, the three central metatarsal rudiments were dissected from E15.5 mice and cultured in αMEM containing 5 µg/ml ascorbic acid, 1 mM β-glycerophosphate pentahydrate, 100 units/ml penicillin, 100 µg/ml streptomycin and 0.2% bovine serum albumin (fatty acid free). The explants were analyzed under a photomicroscope (BZ-X710, Keyence, Japan) for size measurements using Fiji/ImageJ software.

## Histological analysis

For histological analysis, cultured bones were fixed in 4% paraformaldehyde, embedded in Super Cryoembedding Medium (Section-lab, Japan), and frozen in liquid nitrogen. Serial cryosections (6 µm in thickness) were prepared from the fixed specimens and stained with hematoxylin and eosin. In the sectional images, round, columnar, and hypertrophic chondrocytes were distinguished by their characteristic morphological features. Microscopic images were quantitatively analyzed using Fiji/ImageJ software.

## Gene expression analysis

Quantitative RT-PCR analysis was performed as previously described (*Zhao et al., 2016*). From femoral epiphyses, the terminal region containing round chondrocytes and the adjacent region enriched with columnar and hypertrophic chondrocytes were separated under stereo-microscope. Femoral and humeral speciemens were subjected to total RNA preparation using a commercial reagent (Isogen) for cDNA synthesis using a commercial kit (ReverTra ACE qPCR-RT kit). The resulting cDNAs were examined by real-time PCR (LightCycler 480 II, Roche), and the cycle threshold was determined from the amplification curve as an index for relative mRNA content in each reaction.

## Quantification and statistical analysis

All data obtained are presented as the means ± standard error of the mean with *n* values indicating the number of examined mice. Student *t*-test and analysis of variance were used for two-group and multiple group comparisons, respectively (Prism 7, GraphPad Software Inc): $p < 0.05$ was considered to be statistically significant.

## Acknowledgements

We thank Jun Matsushita (Graduate School of Pharmaceutical Sciences, Kyoto University) for mouse in vitro fertilization. This work was supported in part by the MEXT/JSPS (KAKENHI Grant Number 21H02663, 20H03802, and 21K19565), Platform Project for Supporting Drug Discovery and Life Science Research (JP19am0101092j0003), Takeda Science Foundation, Kobayashi International Scholarship Foundation, the NAKATOMI Foundation, Vehicle Racing Commemorative Foundation, The Mother and Child Health Foundation, and Japan Foundation for Applied Enzymology. Y.M. is grateful for Fujita Jinsei Scholarship from Graduate School of Pharmaceutical Sciences, Kyoto University. F.L. is grateful for Scholarship from Graduate Program for Medical Innovation, Kyoto University, and Otsuka Toshimi Scholarship Foundation.

## Additional information

### Funding

| Funder | Grant reference number | Author |
|---|---|---|
| Japan Society for the Promotion of Science | 21H02663 | Hiroshi Takeshima |
| Japan Society for the Promotion of Science | 20H03802 | Atsuhiko Ichimura |
| Japan Society for the Promotion of Science | 21K19565 | Atsuhiko Ichimura |

| Funder | Grant reference number | Author |
|---|---|---|
| Japan Agency for Medical Research and Development | JP19am0101092j0003 | Hiroshi Takeshima |
| Takeda Medical Research Foundation | | Atsuhiko Ichimura |
| Kobayashi International Scholarship Foundation | | Atsuhiko Ichimura |
| Nakatomi Foundation | | Atsuhiko Ichimura |
| Vehicle Racing Commemorative Foundation | | Hiroshi Takeshima |
| Mother and Child Health Foundation | | Atsuhiko Ichimura |

The funders had no role in study design, data collection, and interpretation, or the decision to submit the work for publication.

## Author contributions

Yuu Miyazaki, Data curation, Formal analysis, Investigation, Methodology, Validation, Visualization, Writing – original draft, Writing – review and editing; Atsuhiko Ichimura, Conceptualization, Data curation, Formal analysis, Funding acquisition, Investigation, Project administration, Resources, Validation, Visualization, Writing – original draft, Writing – review and editing; Ryo Kitayama, Naoki Okamoto, Tomoki Yasue, Feng Liu, Takaaki Kawabe, Hiroki Nagatomo, Data curation, Investigation, Methodology; Yohei Ueda, Data curation, Funding acquisition, Investigation, Methodology, Resources; Ichiro Yamauchi, Conceptualization, Data curation, Investigation, Resources; Takuro Hakata, Kazumasa Nakao, Investigation, Resources; Sho Kakizawa, Data curation, Investigation; Miyuki Nishi, Data curation, Investigation, Validation; Yasuo Mori, Haruhiko Akiyama, Resources, Validation; Kazuwa Nakao, Conceptualization, Resources, Validation; Hiroshi Takeshima, Conceptualization, Data curation, Funding acquisition, Investigation, Project administration, Resources, Supervision, Validation, Visualization, Writing – original draft, Writing – review and editing

## Author ORCIDs

Atsuhiko Ichimura (ID) http://orcid.org/0000-0003-0366-5211
Hiroshi Takeshima (ID) http://orcid.org/0000-0003-4525-3725

## Ethics

All experiments in this study were conducted with the approval of the Animal Research Committee according to the regulations on animal experimentation at Kyoto University.

## Decision letter and Author response

Decision letter https://doi.org/10.7554/eLife.71931.sa1
Author response https://doi.org/10.7554/eLife.71931.sa2

# Additional files

## Supplementary files

• Transparent reporting form

## Data availability

All data generated or analysed during this study are included in the manuscript and supporting files. Source data files have been provided for Figures 1, 2, 3, 4, 5, 6, 7 and 8.

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
