## [Editor Report]

With the new additional data and descriptions, the paper in its current state is well organized and data presented add a new information on the role of C-type natriuretic peptide and how it facilitates autonomic Ca^2+^ entry in chondrocytes and modulates bone growth.

---

## [Decision Letter]

**Decision letter after peer review:**

Thank you for submitting your article "C-type natriuretic peptide facilitates autonomic ca^2+^ entry in growth plate chondrocytes for stimulating bone growth" for consideration by *eLife*. Your article has been reviewed by 2 peer reviewers, including Fayez Safadi as Reviewing Editor and Reviewer #1, and the evaluation has been overseen by Mone Zaidi as the Senior Editor. The following individual involved in review of your submission has agreed to reveal their identity: Nazir M Khan (Reviewer #2).

Essential revisions:

1) The authors needs to clarify the definition of "round" chondrocytes. Does this mean differentiating/hypertrophic chondrocytes. If this is the case, then I would replace the term"round" by differentiating or hypertrophic chondrocytes.

2) It would be helpful in the authors include a definitive evidence on the state of the chondrocytes assessed in the study. On other word, please provide a staining or expression of markers for chondrocytes differentiated state, e.g. Sox-9, Collagen X, expression.

3) Data presented on ex vivo bone growth would be strongly supported if histological images are provided (staining of H/E) to define at what stage of chondrocyte differentiation is pharmacological treatment alters chondrocytes differentiation and function.

4) It would be very helpful in the authors corellate the role of C-type natriuretic peptide in endothelial cells and bone growth and how the role C-type natriuretic peptide might contribute to endochondral ossification.

5) Does the role of C-type natriuretic peptide regulate intramembranous bone formation, i.e. does C-type natriuretic peptide regulate osteoblast differentiation/function at a similar fashion as chondrocytes. Please discuss this in the "Discussion section"

6) How does Trpm7 modulate the CNP mediated bone growth? What effect Trpm7 have on on osteogenesis or osteoclastogenesis?

---

## [Author Response]

Essential revisions:1) The authors needs to clarify the definition of "round" chondrocytes. Does this mean differentiating/hypertrophic chondrocytes. If this is the case, then I would replace the term"round" by differentiating or hypertrophic chondrocytes.

As described in many original and review articles [for example, 1, 2], growth plate chondrocytes are morphologically classified into three distinct types; round, columnar, and hypertrophic chondrocytes. We followed the established nomenclature, and “round chondrocyte” is a common word in developmental bone biology. As shown in our previous report, fluorescence microscopic observations can clearly distinguish round, columnar, and hypertrophic chondrocytes in longitudinal bone slices loaded with the ca^2+^ indicators (Qian N., et al., *Sci Signal.*, 12:eaaw4847, 2019). By using the slice preparations, we have previously demonstrated that both round and columnar chondrocytes generate spontaneous TRPM7-mediated ca^2+^ fluctuations that promote self-maturation and growth plate development [3]. We have briefly explained cells assessed in the Method section of the revised manuscript (line 309-311 and line 361-364).

2) It would be helpful in the authors include a definitive evidence on the state of the chondrocytes assessed in the study. On other word, please provide a staining or expression of markers for chondrocytes differentiated state, e.g. Sox-9, Collagen X, expression.

In response to the comment, we examined the expression of several chondrocyte marker genes, including *Sox9*, *Col10a1* and *Runx2*. Previous studies [4-6] indicate that *Col10a1* is exclusively expressed in hypertrophic chondrocytes, whereas *Sox9* expression is mainly observed in both round and columnar chondrocytes. We isolated the terminal region containing round chondrocytes and the adjacent region that is enriched with columnar and hypertrophic chondrocytes from femoral epiphyses. Both the preparations were subjected to total RNA preparations for quantitative real-time PCR analysis. In consistent with the previous reports, *Col10a1* expression was densely observed in the columnar and hypertrophic preparations, whereas *Sox9* expression was similar between both the preparations (Author response image 1). The new data obtained are now included in Figure 1—figure supplement 2 in the revised manuscript.

**Author response image 1. sa2fig1:** Total RNAs were prepared from growth plate sections packed with round chondrocytes or enriched with columnar and hypertrophic chondrocytes from wild-type mice and subjected to quantitative RT-PCR analysis. The data represent the mean ± SEM, and the numbers of mice examined are shown in parentheses. Significant differences between the growth plate sections are marked with asterisks (**p*<0.05 and ***p*<0.01 in *t*-test).

We also analyzed *Col10a1* and *Sox9* expression in chondrocyte-specific *Npr2*-knockout humeral bones. The clarified profiles for the marker genes were largely similar between chondrocyte-specific *Npr2*-knockout and control bones, suggesting that the *Npr2* deficiency does not affect fundamental chondrocytic differentiation in growth plates. The new data obtained are now included in Figure 1—figure supplement 1D in the revised manuscript.

3) Data presented on ex vivo bone growth would be strongly supported if histological images are provided (staining of H/E) to define at what stage of chondrocyte differentiation is pharmacological treatment alters chondrocytes differentiation and function.

In response to the comment, we histologically analyzed cultured CNP-treated bones from the chondrocyte-specific *Trpm7*-knockout (*Trpm7*^fl/fl^, *11Enh-Cre*^+/−^) and control (*Trpm7*^fl/fl^, *11Enh-Cre*^−/−^) mice. The new data obtained clearly indicate that CNP treatments extend the columnar chondrocytic zones in control bones but not in chondrocyte-specific *Trpm7*-knockout bones (Figure 7 in the revised manuscript). We further analyzed in detail CNP-treated metatarsal bones from wild-type mice. CNP-treatments consistently expanded the columnar chondrocytic zones but did not affect the cell densities (Figure 7—figure supplement 1 in the revised manuscript). The expansion seemed to be mainly caused by enlarged extracellular matrix area, although CNP significantly dilated the cell size in columnar and hypertrophic chondrocytes. The observations suggest that CNP promotes cell growth and extracellular matrix production but has no obvious effects on cell proliferation. These results are described in the Results and Discussion sections of the revised manuscript (line 211-214, line 219-220 and line 255-262).

4) It would be very helpful in the authors corellate the role of C-type natriuretic peptide in endothelial cells and bone growth and how the role C-type natriuretic peptide might contribute to endochondral ossification.

Accumulating evidence so-far indicates that CNP functions as autocrine and paracrine factors in several tissues, and is a paracrine vasodilator in blood vessels [7]. Indeed, endothelium-specific *Nppc-*knockout mice exhibit elevated blood pressure due to the deficiency of CNP-induced vasodilation signaling (*Nppc*, the gene symbol of CNP). However, endothelium-specific *Nppc*-knockout mice maintain normal skeletal development [8], indicating that CNP derived from endothelial cells does not correlate with bone growth.

As in our reply to the comment (3), CNP-induced effects became clear in growth plates. To further analyze the effects of CNP on osteoblasts and osteoclasts in developing bones, we performed histochemical analysis as described below (see our reply to the comment 5).

5) Does the role of C-type natriuretic peptide regulate intramembranous bone formation, i.e. does C-type natriuretic peptide regulate osteoblast differentiation/function at a similar fashion as chondrocytes. Please discuss this in the "Discussion section"

In our Kossa-staining analysis of chondrocyte-specific *Npr2*-knockout (*Npr2*^fl/fl^, *Col2a1-Cre*^+/−^) and control (*Npr2*^fl/fl^, *Col2a1-Cre*^−/−^) mice, femoral bones from the E17.5 knockout mice exhibited insufficient mineralization (Author response image 2). In the chondrocyte-specific *Npr2*-knockout bones, the impaired ossification may be indirectly caused by impaired growth plate development. Alternatively, it might be possible that the poor mineralization underlies hyperactivation of osteoclasts and/or hypoactivation of osteoblasts due to non-specific expression of Cre recombinase in the chondrocyte-specific *Npr2*-knockout bones. Although NPR2 expression has not been reported in osteoclasts and osteoblasts, a recent study implies that CNP may directly stimulate osteogenic differentiation in primary cultured osteoblast progenitors [9]. On the other hand, previous in vivo study using *Nppc*-knockout and transgenic mice demonstrates that CNP generally induces the overgrowth of long-bones formed through endochondral ossification, but that CNP does not affect skull bone formed through intramembranous ossification [10], suggesting that CNP has no effect on osteoblastgenesis. Therefore, CNP-induced effects on osteoblasts and osteoclasts have not yet been clarified.

**Author response image 2. sa2fig2:** Impaired bone mineralization in the chondrocyte-specific *Npr2*-knockout embryos. Kossa-stained mid-cross sections of femoral bones from the chondrocyte-specific *Npr2*-knockout (*Npr2*^fl/fl^*, Col2a1-Cre*^+/-^) and control (*Npr2*^fl/fl^*, Col2a1-Cre*^-/-^) E17.5 embryos. Scale bar, 0.3 mm. Both the cross-sectional area and Kossa-positive area were determined from digitalized images, and the Kossa-positive fraction in the cross-sectional area (Kossa-stained ratio) was calculated (graphs). *n* values represent the numbers of mice examined and are shown in parentheses. Significant differences between the groups are marked with asterisks (**p*< 0.05, ***p*<0.01 in *t*-test).

In our previous study, we examined primary cultured osteoblasts and osteoclasts in ca^2+^ imaging, but could not detect spontaneous ca^2+^ fluctuations [11]. Therefore, it is unlikely that the CNP-NPR2-BK channel-TRPM7 channel axis is functioning in both the cell types. To clarify CNP-induced effects on osteoblasts and osteoclasts, it is needed to produce specific model animals; for example, osteoblast-specific *Npr2*-knockout mice using the *Col1a1-Cre* transgene and osteoclast-specific *Npr2*-knockout mice using the *Lyz2-Cre* transgene. Although we are willing to analyze CNP effects on osteoblast/clast in the future project, we cannot unfortunately make meaningful discussion on osteoblasts at the present and regret so much about it.

6) How does Trpm7 modulate the CNP mediated bone growth? What effect Trpm7 have on on osteogenesis or osteoclastogenesis?

We previously reported that TRPM7 channels mediate intermissive ca^2+^ influx in both round and columnar chondrocytes, leading to activation of ca^2+^/calmodulin-dependent protein kinase II (CaMKII) for promoting bone outgrowth [3]. Our preset study demonstrates that CNP-facilitated bone outgrowth essentially requires the activation of TRPM7-mediated ca^2+^ influx. In response to the comment, we analyzed femoral bones from chondrocyte-specific *Trpm7*-knockout mice by conventional activity staining for alkaline phosphatase (ALP, osteoblast marker) and tartrate-resistant acid phosphatase (TRAP, osteoclast maker). The mutant and control bones exhibited similar regional ALP and TRAP-staining densities and distributions (Author response image 3). Although *Trpm7* expression and function are unknown in osteoblasts and osteoclasts at the present, *Trpm7* expression in growth plate chondrocytes seems to have no obvious effects on osteoblastgenesis and osteoclastogenesis.

**Author response image 3. sa2fig3:** Histological analysis in femoral bone of chondrocyte-specific *Trpm7*-knockout mice. Histological analysis of osteoblasts (ALP staining, A) and osteoclasts (TRAP staining, B) in longitudinal sections of femurs from the chondrocyte-specific *Trpm7*-knockout (*Trpm7*^fl/fl^*, 11Enh-Cre*^+/-^) and control (*Trpm7*^fl/fl^*, 11Enh-Cre*^-/-^) E15.5 embryos. Higher-magnification views are also shown in the lower panels. Scale bar, 0.3 mm.

References

1. Hallett, S.A., Ono, W., and Ono, N. (2019). Growth Plate Chondrocytes: Skeletal Development, Growth and Beyond. International Journal of Molecular Sciences *20*, 6009.

2. Jochmann, K., Bachvarova, V., and Vortkamp, A. (2014). Heparan sulfate as a regulator of endochondral ossification and osteochondroma development. Matrix Biology *34*, 55-63.

3. Qian, N., Ichimura, A., Takei, D., Sakaguchi, R., Kitani, A., Nagaoka, R., Tomizawa, M., Miyazaki, Y., Miyachi, H., Numata, T., et al. (2019). TRPM7 channels mediate spontaneous ca^2+^ fluctuations in growth plate chondrocytes that promote bone development Sci Signal. *12*, eaaw4847.

4. Hino, K., Saito, A., Kido, M., Kanemoto, S., Asada, R., Takai, T., Cui, M., Cui, X., and Imaizumi, K. (2014). Master Regulator for Chondrogenesis, *Sox9*, Regulates Transcriptional Activation of the Endoplasmic Reticulum Stress Transducer BBF2H7/CREB3L2 in Chondrocytes. Journal of Biological Chemistry *289*, 13810-13820.

5. Kato, K., Bhattaram, P., Penzo‐Méndez, A., Gadi, A., and Lefebvre, V. (2015). SOXC Transcription Factors Induce Cartilage Growth Plate Formation in Mouse Embryos by Promoting Noncanonical WNT Signaling. Journal of Bone and Mineral Research *30*, 1560-1571.

6. Leung, V.Y.L., Gao, B., Leung, K.K.H., Melhado, I.G., Wynn, S.L., Au, T.Y.K., Dung, N.W.F., Lau, J.Y.B., Mak, A.C.Y., Chan, D., et al. (2011). *SOX9* Governs Differentiation Stage-Specific Gene Expression in Growth Plate Chondrocytes via Direct Concomitant Transactivation and Repression. PLoS Genetics *7*, e1002356.

7. Moyes, A.J., and Hobbs, A.J. (2019). C-Type Natriuretic Peptide: A Multifaceted Paracrine Regulator in the Heart and Vasculature. International Journal of Molecular Sciences *20*, 2281.

8. Nakao, K., Kuwahara, K., Nishikimi, T., Nakagawa, Y., Kinoshita, H., Minami, T., Kuwabara, Y., Yamada, C., Yamada, Y., Tokudome, T., et al. (2017). Endothelium-Derived C-Type Natriuretic Peptide Contributes to Blood Pressure Regulation by Maintaining Endothelial Integrity. Hypertension *69*, 286-296.

9. Watanabe-Takano, H., Ochi, H., Chiba, A., Matsuo, A., Kanai, Y., Fukuhara, S., Ito, N., Sako, K., Miyazaki, T., Tainaka, K., et al. (2021). Mechanical load regulates bone growth via periosteal Osteocrin. Cell Reports *36*, 109380.

10. Nakao, K., Okubo, Y., Yasoda, A., Koyama, N., Osawa, K., Isobe, Y., Kondo, E., Fujii, T., Miura, M., Nakao, K., et al. (2012). The Effects of C-type Natriuretic Peptide on Craniofacial Skeletogenesis. Journal of Dental Research *92*, 58-64.

11. Chengzhu, Z., Ichimura, A., Qian, N., Iida, T., Yamazaki, D., Noma, N., Asagiri, M., Yamamoto, K., Komazaki, S., Sato, C., et al. (2016). Mice lacking the intracellular cation channel TRIC-B have compromised collagen production and impaired bone mineralization. Sci Signal. *9*, ra49.